# Mapping the brain atrophy mediating increased impatience for reward in frontotemporal dementia

Valérie Godefroy [1,2] ✉, Anaïs Durand[3], Richard Levy[3], Bénédicte Batrancourt[3], Liane Schmidt [3], Leonie Koban [1,5] & Hilke Plassmann[3,4,5]

Choices involving trade-offs between larger later (LL) and smaller sooner (SS) rewards—known as intertemporal preferences—are altered in many psychiatric and neurodegenerative conditions, leading to a preference for immediate rewards. Behavioural variant frontotemporal dementia (bvFTD), a neurodegenerative disorder characterised by high impulsivity and atrophies in brain systems relevant for decision-making, provides a neuropathological model to investigate structural networks linked to higher impatience for reward. We studied 22 bvFTD patients and 17 controls, using two intertemporal choice (ITC) tasks involving (1) monetary and (2) food rewards. We compared outcomes of these tasks (discount rate and sensitivity to LL reward amounts) between groups and examined correlations with bvFTD symptoms. We applied whole-brain mediation analysis to participants' structural MRI data to identify neural mediators of higher impatience for reward in bvFTD. BvFTD patients showed higher discount rates and lower sensitivity to LL reward for both money and food. These ITC outcomes for money (but not food) were related to inhibition deficits and lower executive functions among patients. Reduced grey matter density in the medial pulvinar and parahippocampal cortex mediated bvFTD's alteration of ITC outcomes. Lesions of these structures involved in emotional salience and projection may constitute neural markers of impatience for reward.

The degree to which a patient displays impatience in reward-related decision-making can be approximated by measuring 'delay discounting' or 'intertemporal preferences', a methodological approach from behavioural economics. Delay discounting refers to the extent to which people prefer smaller yet sooner over larger but later rewards[1,2]. Delay discounting thus captures the extent to which people focus on the short-term over the long-term consequences of their decisions. Individual differences in people's tendency to discount delayed rewards can be captured by models from behavioural economics, such as the hyperbolic discounting model[2–4], which allows measurement of the discount rate k. Higher discount rates correspond to higher levels of impatience for rewards or higher impulsivity. High discount rates have been detected in a number of psychiatric conditions, such as major depressive disorder, borderline personality disorder, bipolar disorder, bulimia nervosa and binge-eating disorder (see Amlung et al.[5] for a meta-analysis). A growing body of literature has also suggested altered intertemporal preferences and, in particular, increased discount rates in

neurodegenerative conditions such as Parkinson's disease, Alzheimer's disease and frontotemporal dementia (see Godefroy et al.[6] for a review). Given that patients with behavioural variant frontotemporal dementia (bvFTD) show high impulsivity and significant atrophy in brain regions associated with value-based decision-making[7,8], they represent an effective neuropathological model for studying individuals with modified intertemporal preferences. However, no previous study has leveraged the specific pattern of atrophy and the pronounced structural variability within the reward valuation network observed in bvFTD patients to better understand the clinical correlates of reward-related impatience. Beyond replicating previous findings of increased preference for short-term rewards in bvFTD patients (as compared to controls), the goals of the present study were to investigate (1) the clinical symptoms associated with this heightened impatience for reward, and (2) the precise neuroanatomical distribution of damage underlying this behaviour.

[1]CRNL U1028 UMR5292 , Université Claude Bernard Lyon 1, CNRS, INSERM, Centre de Recherche en Neurosciences de Lyon , Bron, France. [2]Developmental Ethology and Cognitive Psychology Lab , Centre des Sciences du Goût et de l'Alimentation, Université Bourgogne Europe, CNRS, Inrae, AgroSup Dijon, Dijon, France. [3]Paris Brain Institute (ICM), INSERM U 1127, CNRS UMR 7225, Sorbonne University, Paris, France. [4]Marketing Area, INSEAD, Fontainebleau, France. [5]These authors contributed equally: Leonie Koban, Hilke Plassmann. ✉e-mail: vlrgodefroy@gmail.com

A few studies[9–15], mostly in healthy populations, have previously investigated the neuroanatomical correlates of discounting, but their results are conflicting and do not allow to draw clear conclusions. Studies in healthy samples evidenced different brain regions in which structural features were related to delay discounting, including the ventromedial prefrontal cortex[9,12], orbitofrontal cortex[12], anterior temporal regions[10,12], temporoparietal junction[12] and dorsolateral prefrontal cortex[14]. These heterogeneous results may reflect the high heterogeneity of methods used by these studies: different populations (adults or adolescents), different structural features (cortical thickness, cortical complexity or grey matter volume) and different statistical approaches (univariate or multivariate). They may also be due to very low effect sizes and low variability within healthy populations[10], an issue which neuropathological model populations (displaying higher variance of both brain structure and discount rates) can contribute to address.

FTD is the most common of a group of neurological conditions associated with predominant degeneration of the prefrontal and temporal regions[16]. BvFTD is the most common clinical variant, characterised by significant changes in personality and behaviour, including disadvantageous monetary decision-making[17]. The main clinical symptoms observed in bvFTD patients are disinhibition, apathy, loss of empathy, perseverative, stereotyped or compulsive behaviour, eating behaviour changes and executive deficits[8]. Despite their highly negative impact on patient's quality of life and caregiver's well-being[18,19], safe and effective solutions to treat these behavioural symptoms are lacking[20,21]. Higher impatience for reward might be a core common factor of several of these symptoms[6]. Therefore, identifying the neuroanatomical correlates of reward impatience in bvFTD may have translational implications. It could, in particular, inform the treatment of potentially related symptoms that involve the urgency facet of impulsivity, defined as the tendency to act rashly, especially in an emotional context, according to the UPPS (Urgency, Premeditation, Perseverance, Sensation seeking) model of impulsivity[22].

How modified intertemporal choices relate to bvFTD specific symptomatology and atrophy pattern is still unclear. Only a few studies have investigated delay discounting behaviour in bvFTD patients. Results suggest that bvFTD patients are steeper discounters and generally more impulsive than healthy controls[11,15,23–25]. These studies focused in particular on comparing patients with bvFTD to patients with Alzheimer's disease (AD) to test the clinical value of using delay discounting to differentiate between these two conditions. Only two studies used voxel-based morphometry to test correlations between discount rates and grey matter density in bvFTD and AD[11,15]. However, these studies were not designed for the specific purpose of studying delay discounting in bvFTD. One of these studies failed to find significant correlations between brain atrophy and discounting rate probably because they did not manage to evidence between-group differences at the behavioural level[15]. The other study[11] found that AD patients showed increased discount rates compared to healthy controls under the influence of viewing emotionally negative pictures prior to choosing between SS and LL rewards. BvFTD patients discounted delayed rewards more than both AD patients and healthy controls, independently of the emotional context. However, this study pooled AD patients, bvFTD patients and controls for the study of the neuroanatomical correlates of discount rates, which was not adapted to establish a pattern of brain structural changes leading to increased discount rates specifically in bvFTD. Besides, their main finding that increased discount rates were associated with greater bilateral amygdala atrophy was driven by AD patients and not by bvFTD patients.

Against this background, in this paper, we hypothesised that the impatience for rewards or preference for short-term rewards is higher in bvFTD patients compared to neurologically healthy controls. Moreover, we assumed that, as potential markers of impatience for reward in bvFTD, delay discounting outcomes should correlate with symptoms of bvFTD that are related to impulsivity (especially its urgency component): (1) disinhibition (or deficit of inhibition), which can be considered as a preference for the most immediate and prepotent answers across various contexts (e.g. preference for immediate reactions of hostility and aggressiveness when confronted to frustration or irritation), (2) deficits of executive functions, which

include lack of inhibitory and attentional control and (3) eating behaviour changes such as binge eating and preference for sweet foods, which may also correspond to the expression of a preference for immediate rewards. We also predicted that the distribution of brain damage in bvFTD patients could explain their higher impatience for reward. Many behavioural disorders in bvFTD are supposed to be closely associated with neurodegeneration in the orbitofrontal cortex (OFC) and ventromedial prefrontal cortex (vmPFC)[26–29]. More generally, the OFC–basal ganglia loops are associated with diverse neuropsychiatric symptoms, including impulsive and compulsive disorders[30,31]. Building on prior work[6], we hypothesised that differences in impatience for reward between bvFTD patients and healthy controls are due to differences in the neuroanatomy of these OFC–basal ganglia loops involved in value-based decision-making[7].

To test these hypotheses, we collected data capturing (1) intertemporal preferences for monetary and food rewards, (2) bvFTD behavioural symptoms, and (3) structural magnetic resonance imaging data (sMRI) from bvFTD patients and matched healthy controls. We investigated impatience for reward with two intertemporal choice tasks for two different rewards (secondary monetary rewards and primary food rewards) to investigate generalisability across reward domains. As shown in Fig. 1, for each of these two rewards, we estimated two potential markers of impatience for reward: the discount rate (log(k)) and the sensitivity to larger later reward amount (i.e. the extent to which the larger later reward amount impacts the participant's choice, which is assumed to be lower with higher impatience for reward). We compared these delay discounting outcomes between patients and controls, and tested their links with bvFTD symptoms. Finally, using the statistical framework of whole-brain mediation analysis[32,33] applied to structural MRI, we aimed at identifying the brain areas in which structural differences between the patients and healthy controls explained group differences in markers of impatience for reward. The whole-brain mediation framework tests each brain voxel as a potential mediator and can be very useful to uncover large-scale brain patterns explaining the link between a disease status and its associated behavioural impairments[34]. This type of analysis goes beyond the correlational framework used by previous studies of associations between brain structure and delay discounting outcomes. Using mediation analysis, we searched for alterations in whole-brain structure playing a robust causal role in the increased impatience for reward observed in bvFTD[35].

## Results

### Higher discount rate and lower sensitivity to LL reward in bvFTD patients

For monetary rewards, bvFTD participants had a mean log(k) parameter of −2.25 (SD = 0.58; median log(k) = −2.11, corresponding to a $k$ of 0.12; range = [−3.35, −1.73]), and healthy controls showed a mean log(k) parameter of −3.33 (SD = 1.80; median log(k) = −3.05, corresponding to a $k$ of 0.05; range = [−6.58, −1.04]). For food rewards, bvFTD participants had a mean log(k) parameter of −1.63 (SD = 0.46; median log(k) = −1.54, corresponding to a $k$ of 0.21; range = [−2.60, −1.16]), and healthy controls showed a mean log($k$) parameter of −2.40 (SD = 1.09; median log(k) = −2.22, corresponding to a $k$ of 0.11; range = [−4.00, −1.16]). Regarding the complementary metric of sensitivity to LL reward, for monetary rewards, bvFTD participants had a mean sensitivity to LL of 0.04 (SD = 0.09; median = 0.02; range = [−0.15, 0.24]), and healthy controls showed a mean sensitivity to LL of 0.16 (SD = 0.12; median = 0.17; range = [0.00, 0.36]). For food rewards, bvFTD participants had a mean sensitivity to LL of 0.009 (SD = 0.03; median = 0.002; range = [−0.06, 0.07]), and healthy controls showed a mean sensitivity to LL of 0.14 (SD = 0.14; median = 0.12; range = [−0.03, 0.48]).

As a robustness check, we verified that, for the money paradigm, increased discount rate was associated with lower sensitivity to LL reward, even after correcting for group effect ($R = -0.34$, $p = 0.04$, 95% CI = [−0.66, −0.005]) (see Supplementary Fig. 1). For the food paradigm, increased discount rate was also associated with lower sensitivity to LL reward after

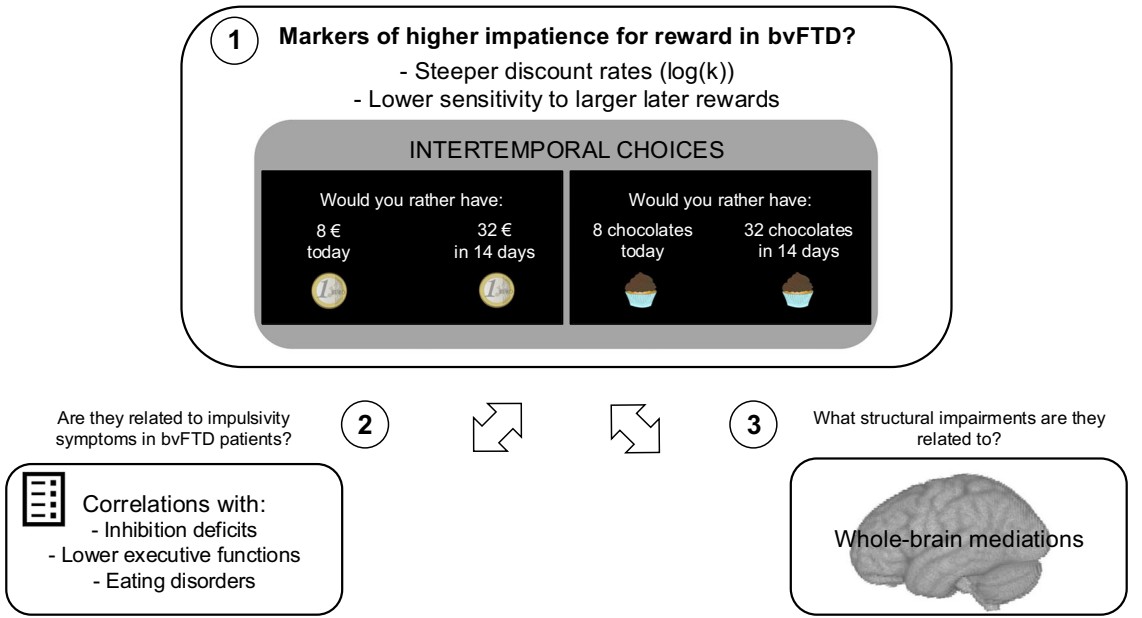

**Fig. 1 | Study objectives and paradigm description.** Study objectives are numbered from 1 to 3. Objective 1 was to identify potential markers of higher impatience for reward in bvFTD patients (compared to controls). For this purpose, we firstly tested the effect of bvFTD on two outcomes of intertemporal choice tasks for money and food rewards: the discount rate and sensitivity to larger later reward amounts. The visual presentation of the two ITC tasks used to assess delay discounting in the study shows an example of trial with monetary rewards on the left and with food rewards (chocolate) on the right. Objective 2 was to further test discount rates and sensitivities to larger later reward as markers of higher impatience for reward in bvFTD by showing their links with impulsivity-related symptoms among patients: (1) inhibition deficits (measured by Hayling test error score); (2) lower executive functions

(assessed by the Frontal Assessment Battery) and 3) eating behaviour changes such as binge eating and preference for sweet foods (measured by the Eating Behaviour Inventory). Objective 3 was to investigate the brain regions in which atrophy may cause higher impatience for reward in bvFTD patients (compared to controls). We used a whole-brain mediation approach applied to participants' grey matter density maps (T1-weighted images pre-processed for VBM) to identify regions mediating the increase in discount rate and the decrease in sensitivity to larger later rewards in bvFTD (compared to controls). The coin and chocolate illustrations are from a public database (https://publicdomainvectors.org/) and are under a CC0 Public Domain Dedication license.

correcting for group effect ($R = -0.55$, $p = 0.001$, 95% CI = [$-0.81$, $-0.18$]) (see Supplementary Fig. 1). These significant links with sensitivities to LL reward confirm that the computed discount rates are related to the tendency to prefer the most immediate options regardless of the offered LL rewards.

BvFTD patients had higher discount rates and thus higher impatience for reward compared to controls for both money rewards ($W = 243$, $p = 0.048$, effect size = 0.32) (see Fig. 2A) and food rewards ($W = 223$, $p = 0.04$, effect size = 0.35) (see Fig. 2B). Relatedly, bvFTD patients showed lower sensitivity to LL rewards than controls for both money ($W = 81.5$, $p = 0.005$, effect size = 0.46) (see Fig. 2C) and food rewards ($W = 61$, $p = 0.003$, effect size = 0.51) (see Fig. 2D). Moreover, we performed two robustness analyses for the comparison of bvFTD patients and controls: (1) As bvFTD and control groups with available ITC data were not matched on age, we checked that differences in discount rates and sensitivity to LL reward between bvFTD patients and controls were still significant while controlling for the effect of age (for discount rates: $B = 0.95$, $p = 0.03$ for money and $B = 0.77$, $p = 0.009$ for food; for sensitivity to LL reward: $B = -0.10$, $p = 0.006$ for money and $B = -0.13$, $p < 0.001$ for food), and (2) we checked that differences in discount rates between bvFTD patients and controls were still present after removing the imputed values of bvFTD patients showing inconsistent patterns of answers, with a close-to-significant effect for money rewards ($W = 186$, $p = 0.07$, effect size = 0.32), and significantly for food rewards ($W = 166$, $p = 0.03$, effect size = 0.41) (see Supplementary Fig. 2).

**Symptom correlates of higher discount rate and lower sensitivity to LL reward in bvFTD patients**

For each of the two ITC task outcomes, we tested correlations with four symptoms suggestive of impulsivity among bvFTD patients: inhibition deficit, lower executive functions, eating behaviour changes (such as binge

eating), and executive apathy (or apathy due to executive dysfunction), for both money and food rewards (leading to a total of 8 tested correlations per ITC outcome). We applied a Bonferroni correction to control for multiple testing of correlations for each ITC outcome.

We found that among bvFTD patients higher lack of inhibition was correlated with higher discount rates for money ($R = 0.67$, $p = 0.0009$, p-corrected=0.007, 95% CI = [0.33, 0.84]) (see Fig. 3A) but not for food ($R = 0.21$, $p = 0.42$, 95% CI = [$-0.38$, 0.66]). Relatedly, higher deficit of inhibition was close-to-significantly linked to lower sensitivity to LL reward for money ($R = -0.41$, $p = 0.07$, p-corrected = 0.56, 95% CI = [$-0.79$, 0.16]) (see Fig. 3C) but not for food ($R = -0.09$, $p = 0.71$, 95% CI = [$-0.54$, 0.35]).

Lower executive functions tended to be related to higher discount rates for money ($R = -0.41$, $p = 0.06$, p-corrected = 0.48, 95% CI = [$-0.75$, 0.005]) (see Fig. 3B) but not for food ($R = 0.02$, $p = 0.94$, 95% CI = [$-0.47$, 0.52]). Lower executive functions were significantly associated with lower sensitivity to LL reward for money ($R = 0.57$, $p = 0.006$, p-corrected = 0.048, 95% CI = [0.20, 0.85]) (see Fig. 3D) but not for food ($R = -0.05$, $p = 0.83$, 95% CI = [$-0.52$, 0.40]).

We did not find any significant correlation between discount rate or sensitivity to LL rewards and eating behaviour changes among bvFTD patients, whether for monetary or for food rewards (although correlation sizes were higher for food rewards—see Supplementary Table 2). However, among bvFTD patients, there was a tendency towards a positive association between the food approach subscale of the EBI and discount rates for food ($R = 0.41$; $p = 0.07$; 95% CI = [$-0.12$, 0.79]). Thus, it is mostly the tendency to rush on food, the disinhibited aspect of eating behaviour changes (or the preference for immediate ingestion of food independent of its taste) that seems to be linked to food reward impatience.

Finally, discount rates and sensitivities to LL rewards for both money and food rewards were not significantly associated with the executive apathy subtype (see Supplementary Table 2), which suggested the specificity of the

**Fig. 2 | Effect of bvFTD on discount rate (log($k$)) and sensitivity to larger later (LL) reward.**
**A** Wilcoxon test of the difference between bvFTD patients ($N = 22$) and controls ($N = 16$) on log($k$) with money rewards; log($k$)-Money was higher in bvFTD patients than in controls ($W = 243$, $p = 0.048$, effect size = 0.32). **B** Wilcoxon test of the difference between bvFTD patients ($N = 21$) and controls ($N = 15$) on log($k$) with food rewards; log($k$)-Food was higher in bvFTD patients than in controls ($W = 223$, $p = 0.04$, effect size = 0.35). **C** Wilcoxon test of the difference between bvFTD patients ($N = 22$) and controls ($N = 16$) on sensitivity to LL reward with money; sensitivity to LL-Money was lower in bvFTD patients than in controls ($W = 81.5$, $p = 0.005$, effect size = 0.46). **D** Wilcoxon test of the difference between bvFTD patients ($N = 21$) and controls ($N = 17$) on sensitivity to LL reward with food; sensitivity to LL-Food was lower in bvFTD patients than in controls ($W = 61$, $p = 0.003$, effect size = 0.51). For each box plot, the lowest horizontal line represents the first quartile (Q1), the middle horizontal line represents the median, and the highest horizontal line is the third quartile (Q3). The lowest end of the box plot is defined as max(min, Q1–1.5*(Q3–Q1)); the highest end corresponds to min(max, Q3 + 1.5*(Q3–Q1)). Black dots correspond to individuals. The coin and chocolate illustrations are from a public database (https://publicdomainvectors.org/) and are under a CC0 Public Domain Dedication license.

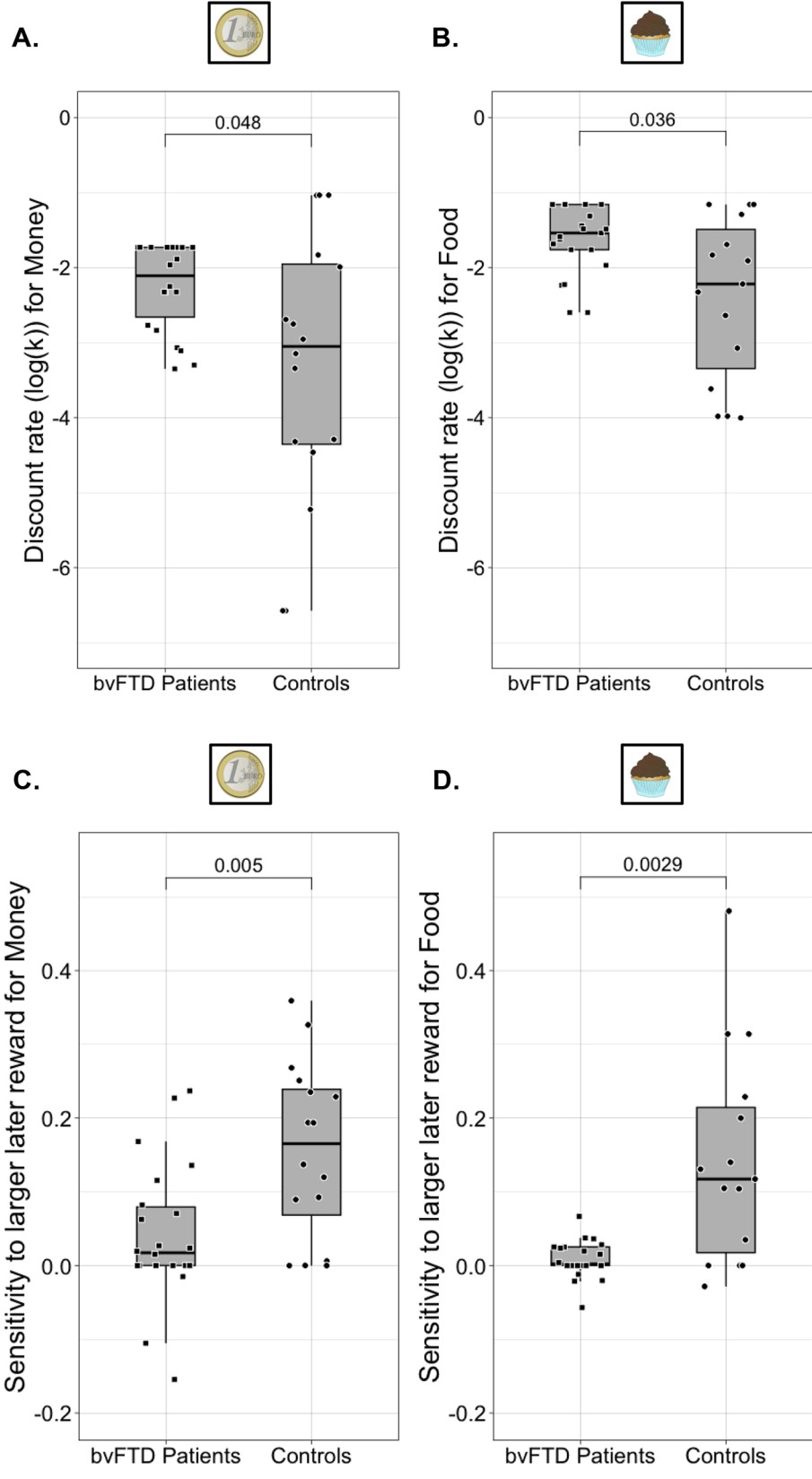

link between ITC outcomes and the impulsivity symptoms of inhibitions deficits and lower executive functions in bvFTD.

Together, results show that the discount rate and sensitivity to LL rewards for monetary rewards can be considered as markers of individual differences of impatience, related to impulsivity symptoms, among bvFTD patients. Among patients, higher discount rates are closely related to inhibition deficits, while lower sensitivities to LL reward are strongly associated with impaired executive functions.

**Fig. 3 | Links between impulsivity symptoms and ITC outcomes among bvFTD patients.**
**A** Spearman correlation between inhibition deficit (measured by Hayling error score) and discount rate for money among bvFTD patients (N = 22; represented as black squares) (R = 0.67, p = 0.0009, p-corrected = 0.007, 95% CI = [0.33, 0.84]); higher inhibition deficit (or preference for immediate prepotent answers) was related to higher log(k)-Money. **B** Spearman correlation between executive functions (measured by FAB score) and discount rate for money among bvFTD patients (N = 22; represented as black squares) (R = −0.41, p = 0.07, p-corrected = 0.56, 95% CI = [−0.79, 0.16]); lower executive functions tended to be related to higher log(k)-Money. **C** Spearman correlation between inhibition deficit (measured by Hayling error score) and sensitivity to LL reward for money among bvFTD patients (N = 22; represented as black squares) (R = −0.41, p = 0.06, p-corrected = 0.48, 95% CI = [−0.75, 0.005]); higher inhibition deficit (or preference for immediate prepotent answers) tended to relate to lower sensitivity to LL-Money. **D** Spearman correlation between executive functions (measured by FAB score) and sensitivity to LL reward for money among bvFTD patients (N = 22; represented as black squares) (R = 0.57, p = 0.006, p-corrected = 0.048, 95% CI = [0.20, 0.85]); lower executive function was related to lower sensitivity to LL-Money.

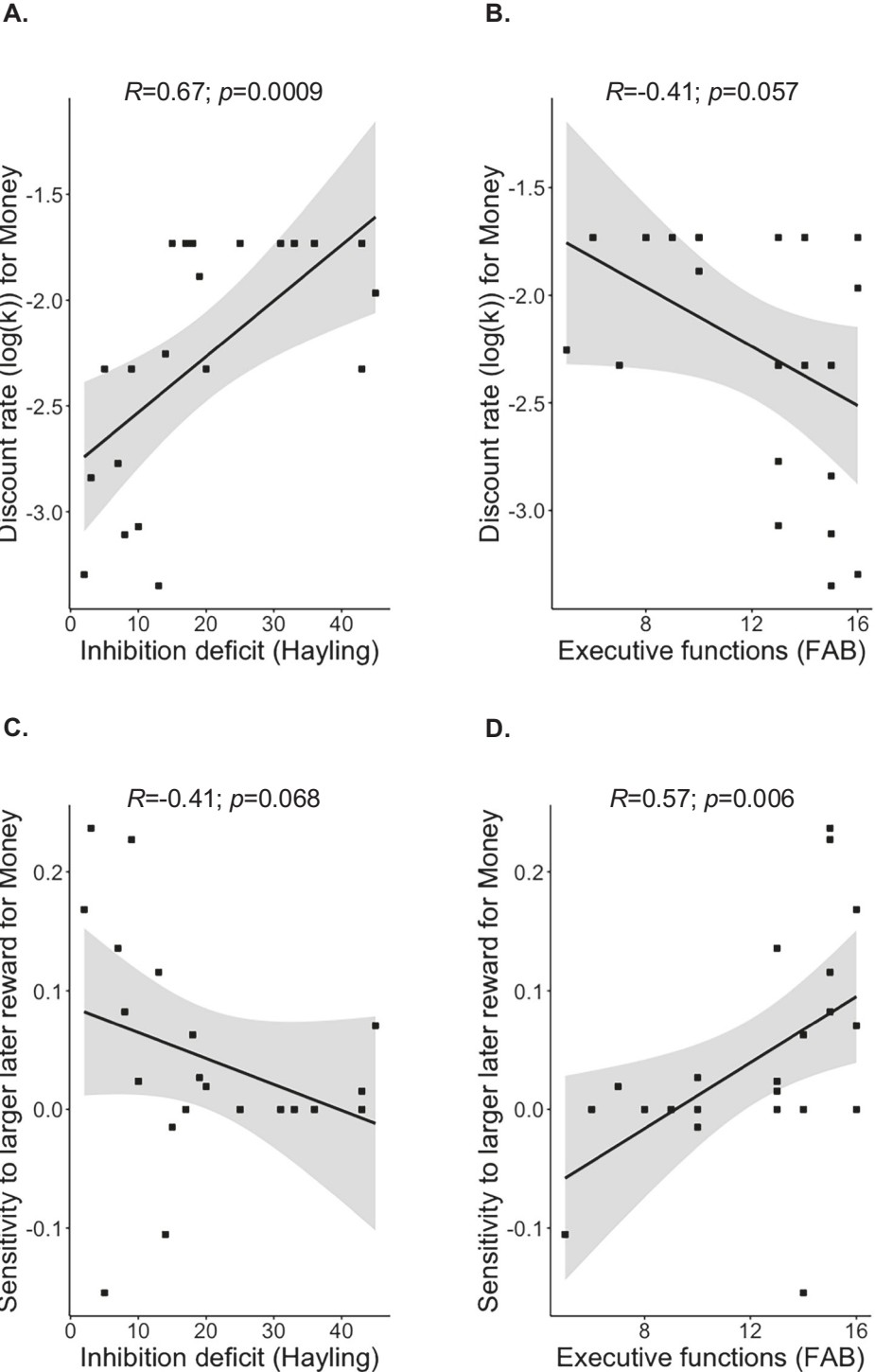

### Brain regions contributing to higher discount rate and lower sensitivity to LL reward in bvFTD patients

Mass-univariate mediation analyses of MRI data estimate independent mediation models for each voxel[32,33]. We identified brain regions involved in three statistical paths (FDR-corrected, q < 0.05, across paths a, b and ab, detailed in next paragraph) of models testing whole-brain mediations for the effect of bvFTD on two ITC outcomes: the discount rate for Money (see Fig. 4) and the sensitivity to LL reward for Money (see Fig. 5). These two outcomes were selected for their previously demonstrated relevance as markers of impatience and impulsivity among bvFTD patients.

As expected, path a, which characterises the effect of bvFTD on brain GMD, showed a very broad pattern of atrophy in bvFTD, including frontal regions (OFC, vmPFC, anterior cingulate cortex, dorsomedial PFC, dorsolateral and lateral PFC, insula), temporal regions (mostly anterior temporal) and subcortical regions (striatum, amygdalae, thalamus, hippocampus). Path b shows the pattern of regions in which GMD predicts the discount rate or the sensitivity to LL reward (for monetary rewards), controlling for the effect of participant group. For path b, we were interested in negative contributors for discount rate (i.e. lower GMD in these regions was related to higher discount rate) and positive contributors for

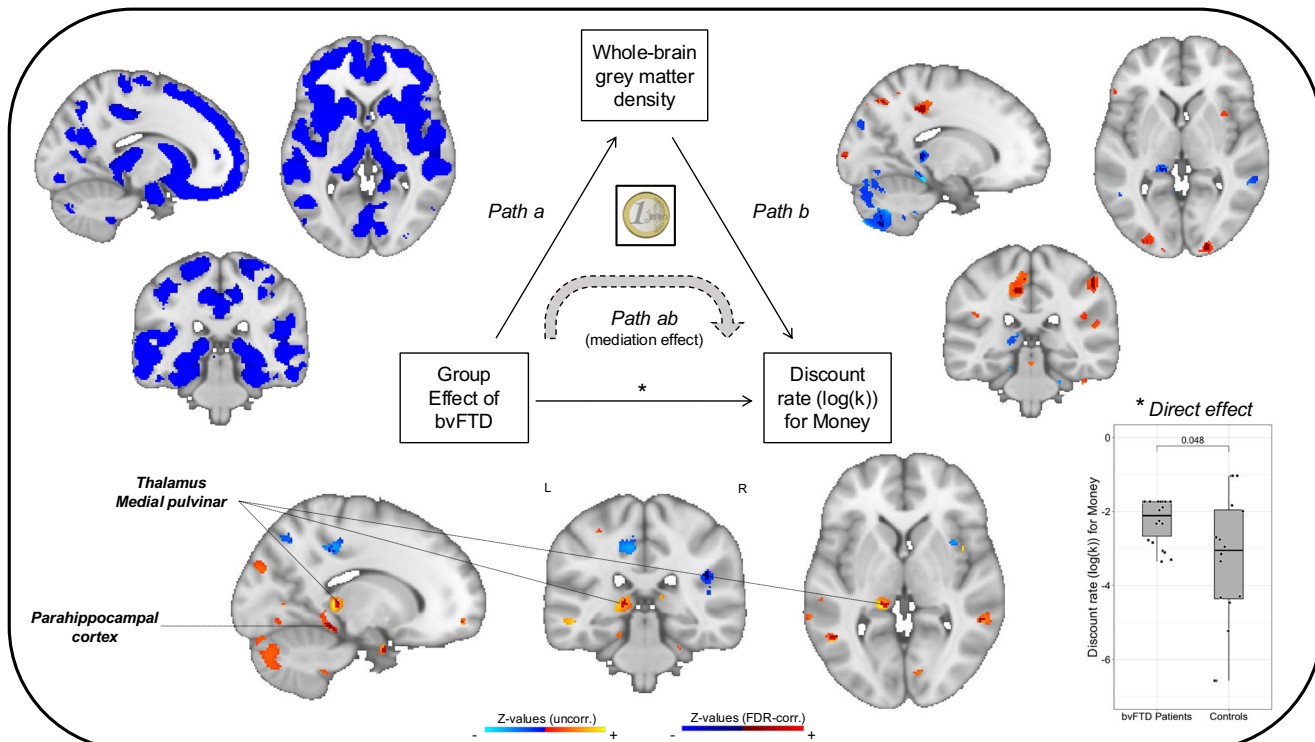

**Fig. 4 | Neuroanatomical whole-brain mediators for the effect of bvFTD on discounting of monetary rewards.** The path diagram displays the voxels of grey matter density that significantly contributed to each path of the mediation model. The blue voxels in *path a* reflect the decreased GMD in bvFTD ($n = 22$) compared to healthy controls ($n = 16$), which corresponds to the typical atrophy pattern observed in bvFTD. *Path b* shows the voxels in which GMD is correlated with the discount rate across all participants ($n = 38$) after controlling for the effect of group. Voxels in blue correspond to a negative correlation in which lower grey matter density contributes to higher discounting. *Path ab* represents brain regions in which GMD mediates the effect of bvFTD on discount rates. Voxels in orange-red correspond to positive mediators that contribute to the increase in discounting due to bvFTD. For *path a*, displayed results are FDR-corrected and thresholded at $q < 0.05$; for *paths b* and *ab*, FDR-corrected results (thresholded at $q < 0.05$) in dark red (positive clusters) and dark blue (negative clusters) are superimposed on uncorrected results (thresholded at $p < 0.05$) in orange and lighter blue. Clusters of main interest for the whole-brain mediation test are positive mediators, significant and FDR-corrected across *paths a, b* and *ab*. The graph on the left shows the significant direct effect of bvFTD condition (compared to healthy controls) on the discount rate of money rewards. The coin illustration is from a public database (https://publicdomainvectors.org/) and is under a CC0 Public Domain Dedication license.

sensitivity to LL reward (i.e. lower GMD in these regions was related to lower sensitivity to LL reward). Finally, path ab represents brain GMD mediating the effect of bvFTD condition on discount rate or sensitivity to LL reward (for monetary rewards). For path ab, we looked at positive mediators contributing to bvFTD patients' increase in discount rates (resulting from the conjunction of negative path a and negative path b) and at negative mediators contributing to patients' decrease in sensitivity to LL reward (resulting from the conjunction of negative path a and positive path b).

The intersection of voxels with significant paths a, b, and ab was then interpreted as a set of mediating brain regions (see results of the conjunction analyses showing intersections between paths for discount rate in Supplementary Fig. 3A and for sensitivity to LL reward in Supplementary Fig. 3B). Among these mediating regions, some were detected in similar locations for both the discount rate and the sensitivity to LL reward with money: mostly in the left medial pulvinar thalamic nucleus and to a lesser extent, in the left parahippocampal cortex (see results of the conjunction analysis showing the intersection between mediating brain regions for discount rate and mediating brain regions for sensitivity to LL reward in Supplementary Fig. 3C). In these regions, bvFTD patients presented significant grey matter atrophy (path a), lower GMD was associated with higher discount rate and lower sensitivity to LL reward (path b), and the loss of GMD due to bvFTD contributed to increase the discount rate and decrease the sensitivity to LL reward in patients (path ab). All the clusters identified for paths b and ab are further detailed in Supplementary Table 3 for the discount rate and Supplementary Table 4 for the sensitivity to LL reward.

## Discussion

The delay discounting paradigm provides estimates of impatience for reward (i.e. the tendency to systematically prefer short-term over long-term rewards), which may constitute a common factor to a set of symptoms involving the urgency facet of impulsivity[22]. A better understanding of the neural bases of this impatience could potentially facilitate the development of effective treatments for such symptoms. In this study, we aimed to advance knowledge of the neuroanatomical basis of impatience for reward by investigating the distribution of structural changes underlying its increase in patients with bvFTD, often serving as a neuropathological model for symptoms involving the frontal lobe and/or basal ganglia[36]. We analysed data from two intertemporal choice tasks—one involving monetary rewards and the other involving food rewards—in bvFTD patients and healthy controls to assess their discount rate (i.e. their tendency to discount the value of delayed rewards) and their sensitivity to larger later reward amounts (i.e. the extent to which larger later reward amounts impact their intertemporal preferences). For both types of reward, bvFTD patients showed increased discount rate and decreased sensitivity to larger later reward compared to healthy controls. This finding confirms previous reports regarding monetary rewards[11,15,23–25,37] and demonstrated that this effect generalises across reward types. Among patients, higher levels of disinhibition and disorders of executive functions were respectively associated with increased discount rates and decreased sensitivity to larger later reward for monetary rewards,

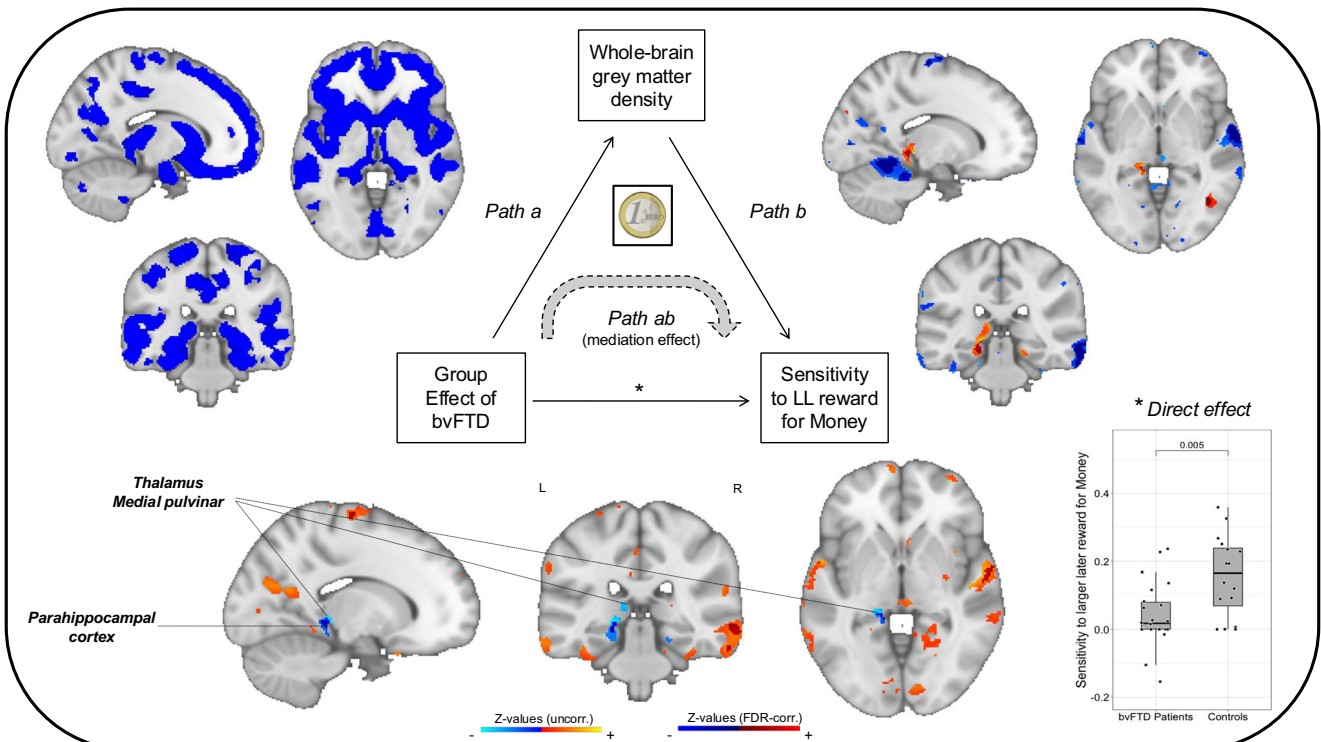

**Fig. 5 | Neuroanatomical whole-brain mediators for the effect of bvFTD on sensitivity to larger later monetary reward.** The path diagram displays the voxels of grey matter density that significantly contributed to each path of the mediation model. The blue voxels in *path a* reflect the decreased GMD in bvFTD (*n* = 22) compared to healthy controls (*n* = 16), which corresponds to the typical atrophy pattern observed in bvFTD. *Path b* shows the voxels in which GMD is correlated with the sensitivity to LL reward across all participants (*n* = 38) after controlling for the effect of group. Voxels in orange-red correspond to a positive correlation in which lower grey matter density contributes to lower sensitivity to LL reward. *Path ab* represents brain regions in which GMD mediates the effect of bvFTD on sensitivity to LL reward. Voxels in blue correspond to negative mediators that contribute to the

decrease in sensitivity to LL reward due to bvFTD. For *path a*, displayed results are FDR-corrected and thresholded at *q* < 0.05; for *paths b* and *ab*, FDR-corrected results (thresholded at *q* < 0.05) in dark red (positive clusters) and dark blue (negative clusters) are superimposed on uncorrected results (thresholded at *p* < 0.05) in orange and lighter blue. Clusters of main interest for the whole-brain mediation test are negative mediators, significant and FDR-corrected across *paths a, b* and *ab*. The graph on the left shows the significant direct effect of bvFTD condition (compared to healthy controls) on the sensitivity to larger, later money rewards. The coin illustration is from a public database (https://publicdomainvectors.org/) and is under a CC0 Public Domain Dedication license.

providing further evidence that these two outcomes constitute markers of impatience for reward in bvFTD. Moreover, atrophy and reduced grey matter density in the thalamus (medial pulvinar nucleus) and in the parahippocampal cortex were found to play a key role in the variation of these two markers of impatience for reward.

We found that higher discount rates were closely connected with a lack of sensitivity to the magnitude of the larger later amounts, in accordance with Frost and McNaughton's theory of delay discounting[38]. This theory suggests that for steeper discounters, delayed rewards are less salient (and therefore less likely to be chosen): for these individuals, reward salience intensely decreases with the perceived mental distance to reach it[38]. Results indeed suggest that bvFTD patients act as if they tended to ignore the less salient later options and were biased in favour of the most salient, smaller, but sooner options, thus confirming previous observations in bvFTD[23].

Higher impatience for reward was a common factor related to two symptoms in bvFTD: inhibition deficits and executive disorders. Inhibition deficits were in particular closely related to higher discount rates of money among patients. In a previous study that used factor analysis to identify impulsivity dimensions in patients with Parkinson's disease, they found a factor of preference for immediacy highly related to both discount rates of money and deficits of inhibition measured by the Hayling test[39]. The reduced attraction of delayed rewards associated with increased discount rates might be part of a more general mechanism of reduced attention to the most distant answers to any stimulation. By causing a decrease in attention towards the semantically distant words, this mechanism may, in particular, explain patients' difficulty in inhibiting the semantically close words in the

Hayling test[40]. Moreover, executive disorders were mostly associated with lower sensitivity to larger, later monetary rewards among patients. Executive functioning has previously been connected with discounting of different outcomes in university students[41]. Executive disorders may impact in particular the processing of trial parameters that are necessary to the anticipation and comparison of option consequences in the ITC task.

The observed links between the ITC outcomes for money and bvFTD symptoms suggestive of urgency and lack of control confirm the validity of these outcomes as markers of impatience for reward in bvFTD. On the opposite, we did not find such links for the ITC outcomes with food rewards among patients. With food stimuli, bvFTD's modified intertemporal preferences might not reflect their general impatience for reward and lack of attention to the most distant options. It might rather reflect their changes in appetite regulation and their modified sensitivity to the specific food rewards associated with hormonal and metabolic changes[42–44]. Indeed, the ITC outcomes for food rewards were more closely associated with eating behaviour changes than outcomes for money rewards, and the discount rate for food was close-to-significantly related to one specific facet of eating behaviour changes (the food approach tendency or tendency to rush on food[45]).

Results of the whole-brain mediation did not evidence regions of the OFC-basal ganglia loops as consistent mediators of altered intertemporal preferences in bvFTD, contrary to what we expected. However, thalamic lesions, which impact the OFC-basal ganglia loops, were found as important contributors to the alteration of both the discount rate of money and the sensitivity to larger, later money rewards in bvFTD. Grey matter density in a

specific nucleus of the thalamus, the medial pulvinar, was indeed evidenced as an important mediator of the increase in discount rate and the decrease in sensitivity to larger later reward due to bvFTD. Frost and McNaughton proposed that the 'crude' positive subjective value of the delayed gain is linked to activity in the thalamus, which is a key node in feedback loops between basal ganglia, hippocampus, and prefrontal cortex[38]. The medial pulvinar is highly connected with nodes of the salience network—an intrinsic connectivity network that functions to represent the emotional significance of internal and external stimuli in order to guide behavioural responses[46–48]—such as the anterior insula and ACC[49]. Damage to the pulvinar has been shown to disrupt functional connectivity in the salience network in bvFTD patients[50]. Atrophy to the medial pulvinar may thus increase impulsive decision-making by causing dysfunctions of the salience network, thereby reducing the positive salience of larger later rewards. The medial pulvinar is also assumed to coordinate the processing of concurrent information and play a key role in determining decisions in conflictual situations with competing constraints, as in the delay discounting task[51,52]. Thus, evidencing the medial pulvinar as mediator of the alteration of intertemporal preferences in bvFTD is consistent with the assumption that the impairment of conflict detection is a central mechanism underlying impatient choices in this condition[6]. Additionally, the medial pulvinar has a significant role in the pathogenesis of bvFTD, especially in carriers of the C9orf72 genetic mutation, who may present medial pulvinar atrophy due to a developmental lesion[50,53]. Abnormal development of the medial pulvinar also constitutes a common factor of several neurodevelopmental diseases involving impulsivity disorders[51,52] that may predispose to developing bvFTD and other neurodegenerative diseases (for reviews, see refs. 54–58). Thus, the role that our findings suggest for medial pulvinar is in line with the idea that the alteration of intertemporal preferences may represent the expression of a common genetic background explaining the likely continuum between neurodevelopmental and neurodegenerative diseases[6]. As such, delay discounting would be a great candidate marker for the very early prediction of a risk of neurodegenerative process.

Overlapping clusters in the parahippocampal cortex were also evidenced as mediators of the modifications of the two markers of impatience for reward in bvFTD. Lower grey matter density in the parahippocampal cortex mediated bvFTD's alteration of discount rate and sensitivity to larger later option for money rewards. The parahippocampal cortex is involved in the consolidation of episodic memory, known to impact delay discounting outcomes[59]. Along with other regions of the medial temporal lobe, it is assumed to contribute to a prospection network that mediates individual differences in intertemporal preferences through its role in simulating future outcomes[60]. Evidencing the parahippocampal cortex as a mediator suggests that higher impatience for reward in bvFTD is not only related to impairments of reward valuation due to lack ok emotional salience; it also involves deficits in the contextualisation of potential rewards.

Several limitations of our study call for future work in this area to provide more evidence for our results and their implications. A first limitation is the relatively small sample size of bvFTD patients, partially due to the requirements of the protocol. BvFTD patients who participated in this study were chosen according to very selective inclusion criteria (e.g. MMSE score >20), because of the need to include patients at a very early stage. The small sample size leads to a substantial lack of statistical power for the mediation analyses, which might undermine the reliability of these results. Our sample size of bvFTD patients is, however, very close to the upper limit of the range of sample sizes used in previous studies of delay discounting in bvFTD (i.e. between $N = 14$ and $N = 28$). Moreover, we could still identify significant clusters with an interesting convergence of results between two different frameworks used to characterise altered intertemporal preferences in bvFTD: the assumption-ladden outcome of discount rate and the less assumption-ladden outcome (not driven by any hypothesis of hyperbolic discounting) of sensitivity to larger later reward. A second limitation of our results is the small size of effects detected for *path ab* (mediation path). We could detect very small effect sizes for *path a* (effect of bvFTD on grey matter density) and only moderate to large effect sizes for *path b* (effect of grey

matter density on ITC outcome after controlling for group effect), which implied that significant results for *path ab* were mostly driven by *path b*. Consequently, clusters evidenced as mediators (in particular, the medial pulvinar and parahippocampal cortex, according to the results of the conjunction analysis between *paths a, b* and *ab*) were also regions for which the effect was independent of bvFTD group effect on brain atrophy. This is supposed to prevent the well-documented co-atrophy issue[61] and suggests a possible transdiagnostic value of our results. However, further elucidation of the neural networks underlying changes in impatience for reward and related symptoms would require a more large-scale study, involving patients with divergent neurodegenerative diseases. A third limitation of this study was the necessity to approximate several discounting values in cases of unique choice patterns (either only SS or only LL options) and cases of very inconsistent choices, especially in bvFTD patients. It is possible that our paradigm was not adapted enough to reflect a very large range of discounting parameters, in particular, very high discount rates in bvFTD. The actual range of discount rates might be larger in bvFTD patients. Of note, even after removing the cases of inconsistent patterns of choice, discount rates were still found higher (at least in tendency) in bvFTD patients than in controls. Additionally, results were further supported by the second ITC outcome of sensitivity to larger, later rewards, which did not require any approximation. Future studies would benefit from the use of paradigms of delay discounting that are more adjusted to their specific patient population.

In conclusion, this study investigated the relationships between markers of impatience for reward, symptoms and brain structure in the neuropathological condition of bvFTD. Key findings suggest that in bvFTD: (1) the preference for smaller sooner money rewards relates to both inhibition deficits, revealing preference for most immediate and salient answers, and to executive disorders that impact the processing of trial information; (2) the increased impatience for reward is caused by structural lesions in specific brain areas, in particular the medial pulvinar and parahippocampal cortex. These findings stress the importance of two types of functional impairments that may play a central role in excess impatience for reward. Firstly, disorders of emotional salience processing may bias reward valuation at the advantage of the most salient immediate option. Secondly, memory and projection issues may be linked to difficulties of information processing, preventing the correct anticipation of consequences of reward options. In terms of translational impacts, our results suggest that the delay discounting task provides markers of inhibition deficits and executive disorders and could thus potentially contribute to a better phenotyping of conditions associated with a marked impatience for reward. They also suggest that the anatomy of specific brain regions, such as the medial pulvinar and parahippocampal cortex, may constitute targets of interest for the early detection and treatment of impulsivity symptoms involving urgency. Cortico-striatal-thalamic loop circuits are known as central pathways and targets for the treatment of many conditions[31] but targeting specific regions within these circuits might be particularly efficient. For instance, novel treatments targeting abnormalities in the medial pulvinar through invasive and non-invasive brain stimulation might have indirect effects to reduce symptoms of impulsivity. However, one of the main limitations of our results is that they only concern bvFTD condition and need to be further validated across a broader panel of neurodegenerative and psychiatric conditions. This study paves the way for future research to disentangle the distinct contributions of brain regions directly involved in valuation, like the OFC, from those more indirectly engaged in valuation, such as the medial pulvinar and parahippocampal cortex, in shaping impulsivity symptoms.

## Methods
### Participants
This study was part of a larger-scale protocol that also collected other data that are reported elsewhere[27,62–64]. Participants were recruited in the context of a clinical study at the Paris Brain Institute in France (Clinicaltrials.gov: NCT03272230)[27,62–64]. This clinical study aimed at assessing several behavioural symptoms (disinhibition and apathy) and investigating their neural correlates in behavioural variant frontotemporal dementia. BvFTD patients

**Table 1 | Demographic and main clinical measures of bvFTD patients and controls**

|  | bvFTD | Controls | bvFTD vs. controls |
|---|---|---|---|
| % women | 36.4% | 52.9% | $\chi^2 = 0.5; p = 0.48$ |
| Age | 66.5 (8.5) | 62.2 (7.2) | $W = 256; p = 0.05$ |
| Age range | [45; 82] | [46; 71] | _ |
| Education level | 6.1 (2.0) | 7.2 (1.1) | $W = 136; p = 0.13$ |
| MMSE (/30) | 23.8 (2.6) | 29.5 (0.7) | $W = 5.5; p < 0.001$ |
| DRS (/144) | 119.4 (8.9) | 142.2 (1.3) | $W = 0; p < 0.001$ |
| Hayling (errors) | 19.8 (13.7) | 3.3 (2.5) | $W = 331.5; p < 0.001$ |
| FAB (/18) | 12.1 (3.4) | 17.4 (0.9) | $W = 6.5; p < 0.001$ |
| EBI (/32) | 13.1 (6.2) | 1.4 (1.9) | $W = 360; p < 0.001$ |
| DAS-Executive (/24) | 10.0 (4.6) | 4.2 (3.6) | $W = 311.5; p < 0.001$ |

Data are given as mean (SD). BvFTD patients: N = 22/controls: N = 17. For comparison, we used Wilcoxon tests for non-normally distributed variables and Student's t-tests for normally distributed variables. First, the main clinical measures are listed.
*MMSE* mini-mental state examination, *DRS* dementia rating scale. Second, variables of interest in the study are presented, *Hayling (errors)* objective measure of inhibition deficit from the Hayling Sentence Completion Test (number of errors in the inhibition phase of the test), *FAB* frontal assessment battery, *EBI* eating behaviour inventory, global measure of changes in eating behaviour, *DAS-Exe* dimensional apathy scale, subscale measuring lack of executive functions to complete goal-directed behaviours.

were recruited in two tertiary referral centres in Paris: the Pitié-Salpêtrière Hospital and the Lariboisière Fernand-Widal Hospital. They were diagnosed according to the International Consensus Diagnostic Criteria[8]. Inclusion criteria for bvFTD patients included presenting a Mini-Mental State Evaluation (MMSE) score of at least 20 to ensure that they would have the ability to undergo the full protocol. Healthy controls were recruited by public announcement; inclusion criteria were a MMSE score superior to 27 and matching the demographic characteristics of the bvFTD patient group (i.e. age, gender and education level). In total, 24 bvFTD patients (mean age = 66.6, 66.6% male) and 18 neurologically healthy controls (mean age = 62.6, 44.4% male) were recruited for the overall clinical study. Data were missing for some participants (total of four missing participants for each of the two intertemporal choice tasks) because of technical issues with the touch tablet used to collect the data of the intertemporal choice tasks. Hence, in this study, we used data from 22 bvFTD patients and 16 healthy controls for delay discounting of money and data from 21 bvFTD patients and 17 healthy controls for delay discounting of foods[65]. For additional details regarding the sample of 22 bvFTD patients and 17 healthy controls with at least one measure of delay discounting, refer to Table 1. Among the 22 bvFTD patients included in the analyses, five patients (22.7% of the patient sample) had a pathological expansion in the C9 open reading frame 72 gene.

This study is part of clinical trial C16-87 sponsored by INSERM, the French national institute for biomedical research. It was granted approval by the local Ethics Committee ('Comité de Protection des Personnes') on 17 May 2017 and registered in a public registry (Clinicaltrials.gov: NCT03272230). All ethical regulations relevant to human research participants were followed. All study participants gave their written informed consent to participate, according to the Declaration of Helsinki and in line with French ethical guidelines.

**Intertemporal choice tasks**

Participants performed two intertemporal choice (ITC) tasks, one using monetary rewards and one using food rewards, in a randomised order. Both types of rewards were matched in economic value, meaning that one chocolate truffle used as a food reward had a retail price of €1. Each of these tasks consisted of 32 choices between smaller sooner and larger later rewards (see Fig. 1). To incentivize participants to give us truthful answers, they were

instructed that one of their 32 choices would be randomly selected at the end of the experiment, and the option they had chosen would be given to them at the indicated time. Thus, participants' choices were incentive-compatible and non-hypothetical. Participants indicated their choice by pressing either a blue key on the keyboard with their right index finger to select the option on the left or a yellow key with their right middle finger to select the option on the right. Once the choice had been made, a message on the screen indicated the selected option. There was a test trial at the beginning of each task to verify participant's correct understanding, and an experimenter was present to address potential questions throughout the task (especially for bvFTD patients). The maximum time between choice trials was either 3.5, 4 or 5 s (randomised order) and depended on the participant's response time for each choice. The values of the SS reward ranged from €8 to €35; the delay for the SS option was either 0, 14 or 28 days. LL options varied between €10 and €96; the delay for the LL option was either 14, 28 or 42 days. For the ITC task with food rewards, choice trial parameters were exactly the same except that reward amounts in euros were replaced by numbers of chocolate truffles with the economic value of €1 each. Supplementary Table 2 details all combinations of SS and LL options and delays that constituted the choice trials of the task, as well as the corresponding indifference $k$'s ($k$'s for which the presented LL and SS options would be chosen with equal probability). For each of the two ITC tasks, choice trials were presented in randomised order.

**Measures of impulsivity-related bvFTD symptoms**

We used measures of three symptoms of bvFTD[8] that we assumed to be associated with impatience for reward: (1) the deficit of inhibition measured by the Hayling Sentence Completion Test (HSCT), (2) lower executive functions measured by the Frontal Assessment Battery (FAB) and (3) eating behaviour changes measured by the Eating Behaviour Inventory (EBI).

The Hayling Sentence Completion Test[40] asks participants to complete 15 sentences using the appropriate word, as fast as possible (automatic condition, part A—e.g. for 'The rich child attended a public ___', the correct answer is 'school'), and 15 sentences using a completely unrelated word (inhibition condition, part B—e.g. for 'London is a very lively ___', 'city' is considered an incorrect answer, but 'banana' would be considered correct). We selected the Hayling error score (number of errors in part B) as a measure of the difficulty in inhibiting a prepotent immediate response, as in Flanagan et al.[66]. Among the different tests available to assess inhibition deficits, the Hayling test presents several advantages: it is an objective measure, and impaired performances are distinctive characteristics of bvFTD patients[67].

The Frontal Assessment Battery (FAB) is a brief tool designed to evaluate executive functions, lower scores indicating worse executive functions[68]. This battery consists of six subtests, each exploring one function: conceptualisation and abstract reasoning, mental flexibility, motor programming and executive control of action, resistance to interference, self-regulation, inhibitory control, and environmental autonomy.

The Eating Behaviour Inventory is a questionnaire with 32 questions investigating recent changes in four domains of eating behaviour: eating habits (e.g. 'Seeks out food between meals'), food preference (e.g. 'Is more attracted by sweet foods'), table manners (e.g. 'Is eager to start eating'), and food approach (e.g. 'Eats closer to his plate')[45]. This questionnaire is particularly adapted to detect the specific eating behaviour changes of bvFTD and has been evidenced as a tool helping the differential diagnosis of this condition. For bvFTD patients, this questionnaire was completed by the caregiver to avoid self-report biases due to anosognosia in patients.

In order to investigate the specificity of the link between intertemporal choice task outcomes and these impulsivity-related symptoms, we also tested their links with apathy, another core symptom of bvFTD. As apathy is known to be a complex multidimensional symptom, we used the Dimensional Apathy Scale (DAS)[69] to measure three subtypes of apathy derived from the theoretical model proposed by Levy and Dubois[70]: initiation apathy (deficit of initiation of thoughts and actions; 8 items; e.g. 'I set goals for myself' as reversed item), emotional apathy (emotional blunting; 8 items;

e.g. 'I become emotional easily when watching something happy or sad on TV' as reversed item), and executive apathy (impairment in executive functions to manage goals; 8 items; e.g. 'I find it difficult to keep my mind on things'). We assumed that intertemporal choice task outcomes could, in particular, be related to executive apathy.

## MRI data acquisition and pre-processing

Brain imaging data were acquired at the neuroimaging centre (CENIR) of the Paris Brain Institute with a Siemens Prisma whole-body 3 T scanner (with a 12-channel head coil). Structural images were acquired using a T1-weighted MPRAGE sequence (TR 2400 ms; TE 2.17 ms; FOV 224 mm; 256 slices; slice thickness 0.70 mm; TI 1000 ms; flip angle 8°; voxel size 0.7 mm isomorphic; total acquisition time 7:38). T1 images were pre-processed for voxel-based morphometry (VBM) analyses using the Statistical Parametric Mapping (SPM) software (version 12). The pre-processing consisted of several standardised steps. Image files were first segmented and registered using rigid linear deformations. These images were then used as input to create a customised DARTEL template and individual flow fields for each subject. The DARTEL module determines the nonlinear deformations for warping all the grey and white matter images so that they match each other. Finally, spatially normalised and smoothed Jacobian scaled grey matter images were generated, using the deformations estimated in the previous step.

## Statistics and reproducibility

The statistical analyses were performed in several steps (see Fig. 1). First, we verified that bvFTD patients were more impatient for smaller, sooner rewards as compared to controls. Second, we tested whether potential markers of impatience for reward (discount rate and sensitivity to larger later reward) correlated with bvFTD symptoms suggestive of impulsivity among patients. Third, we investigated in which brain regions the grey matter atrophy mediated the effect of bvFTD on markers of preference for sooner options. All the analyses were performed using R Studio (1.2.1335) and Matlab (R2017b) toolboxes.

**Computation of discount rates and sensitivities to LL rewards.** For each participant and for each reward type, we first computed his or her discount rate $k$. For each choice, we computed the corresponding theoretical 'indifference' discount rate $k$ (i.e. the theoretical $k$ value for which an individual would consider both options of the trial as equivalent assuming a hyperbolic discount function where $V = A/(1 + kD)$) using the following equation:

$$k = \frac{A_2 - A_1}{(A_1 D_2 - A_2 D_1)}$$

where $A_1$ = absolute value of SS reward, $A_2$ = absolute value of LL reward, $D_1$ = SS delay and $D_2$ = LL delay. We fitted a logistic probability function relating the theoretical indifference $k$ value of a trial to the probability of choosing the LL option. We then used this function in each participant to identify their indifference point, that is, the theoretical indifference $k$ value at which their probability of LL choice was equal to 50% (and thus equal to the probability of SS choice). Because $k$ values are generally not normally distributed, we used the log-transformed values of $k$ ($\log(k)$).

We observed a frequent 'mono-choice' pattern of answer, especially only SS chosen throughout the 32 trials. Seven bvFTD patients exclusively chose the SS option for the monetary task, and five bvFTD patients exclusively chose the SS option for the food task (four of them only chose SS for both tasks). Only two controls exclusively chose the SS option for the monetary task as well as for the food task (one of them only chose SS for both tasks). Note that this pattern of behaviour cannot be explained by a lack of attention since the side of the screen on which the SS option was displayed was counterbalanced. Put differently, participants exhibiting such a mono-choice pattern did actively identify the SS options. Moreover, inconsistent patterns of answers (i.e. with the probability of choosing the LL option

decreasing with larger amounts of LL rewards) were observed, only in bvFTD patients (five for the money paradigm and six for the food paradigm). For these participants showing patterns of unique answers or very incoherent answers, the discounting model did not fit, and we could not use the same method of estimation. We took advantage of the strong linear relationship between %SS choice and estimated $k$'s (for monetary rewards, $R = 0.64$; $p = 0.001$ and for food rewards, $R = 0.69$; $p < 0.001$), and within each population sample (bvFTD and controls), we imputed $k$ values where they were missing. For participants with 0% SS choice, the imputed value was the minimum $k$ observed in the rest of the sample; for participants with 100% SS choice, the imputed value was the maximum $k$. For participants with very inconsistent answers, we used a linear interpolation method to approximate missing $k$'s from %SS values.

For each participant and each reward type, we also computed the individual sensitivity to larger later (LL) reward as an outcome that does not rely on an a priori assumption of hyperbolic discounting, to complement the discount rates. For this purpose, we fitted a logistic regression model in each participant, predicting the trial-to-trial probability of choosing the LL option from the LL reward amount and LL delay value. We used the regression coefficient of LL reward amount as an estimate of the individual sensitivity to LL reward (higher sensitivity to reward corresponding to higher values).

To avoid biasing correlation and mediation analyses, extreme outliers (i.e. <Q1–3*(Q3–Q1) or >Q3 + 3*(Q3–Q1)) identified within the whole sample were systematically removed for both the computed discount rates and sensitivities to larger, later rewards. Thus, two participants (in control group) were removed for the discount rate for food, and three other participants (two in control group and one in patient group) were removed for the sensitivity to LL reward for food. Moreover, for both the money and food paradigms, to check the validity of our imputation method for the computation of discount rates, we verified that the participants' discount rates were related to how sensitive they were to LL reward. Using nonparametric Spearman rank correlations, we tested the links between the computed discount rates and sensitivity to LL reward after correction for group effect to verify that higher discount rates were associated with lower sensitivity to LL reward. More precisely, we tested the correlations between the residuals of the linear regression predicting discount rates from group (bvFTD vs. control) on the one hand and the residuals of the linear regression predicting sensitivity to LL rewards from group (bvFTD vs. control) on the other hand. These residuals account for the part of the variability in the scores of discount rate and sensitivity to LL reward that is not due to group effect.

**Effect of bvFTD on discount rate and sensitivity to LL reward.** Our hypothesis was that higher discount rates would be observed in bvFTD patients compared to controls. We compared discount rates in bvFTD patients and controls using non-parametric Wilcoxon tests. Further, if bvFTD patients are indeed more impatient for reward regardless of the amount offered by the LL option, they should also show lower sensitivity to LL reward. We used nonparametric Wilcoxon tests to compare bvFTD patients and controls on sensitivity to LL reward calculated for money and food rewards.

For a high expected effect size on the difference between bvFTD and controls on delay discounting ($P(X > Y) = 0.76$), the used sample sizes for money and food delay discounting allowed to reach a power close to 80% (with a two-sample Wilcoxon test).

**Symptom correlation with discount rate and sensitivity to LL reward in bvFTD.** Our hypothesis was that symptoms of inhibition deficits, lower executive functions and eating behaviour changes were related to higher discount rate and lower sensitivity to LL reward in bvFTD patients. We used non-parametric Spearman rank correlations to test the links between the participants' discount rates ($\log(k)$) for money and for food rewards, and: (1) the Hayling error score (measuring inhibition deficit), (2) the FAB (measuring executive functions) and (3) the EBI total score (measuring eating behaviour changes). We also tested the links with the executive apathy subtype (not with the initiation and emotional apathy

subtypes) since executive apathy is conceptually related to executive dysfunctions. We tested all these associations within the group of bvFTD patients. The exact same correlation tests were performed with the sensitivity to LL reward for money and food rewards.

**Structural brain mediators of modified discount rate and sensitivity to LL reward in bvFTD.** To investigate which brain regions contribute to the differences in discount rate and sensitivity to LL reward between bvFTD patients and healthy controls, we used whole-brain mediation analyses[32,33] applied to structural MRI data. In this approach, each brain voxel is tested as a potential mediator of the effect of bvFTD on discount rate or sensitivity to LL reward. This approach allowed us to identify regions across the whole brain in which grey matter density (GMD) loss causally explained the effect of bvFTD condition on discount rates and sensitivities to LL reward (i.e. path ab, mediation effect). This analysis also allowed us to identify brain regions in which GMD would be different between the groups (i.e. path a, group effect) and predictive of the discount rate or sensitivity to LL reward, controlling for group effect (i.e. path b, intrinsic link between neuroanatomy and delay discounting). We were interested in brain structures being involved in all three effects.

To this end, participants' GMD maps (T1 images pre-processed for VBM) were submitted to a single-level whole-brain mediation analysis. We used a percentile bootstrapping procedure (with 10,000 generated samples) to detect the most robust mediators. Resulting statistical maps were thresholded at $p < 0.05$ FDR-corrected across the whole brain and across path a, b and ab analyses (corresponding to a voxel level of $p < 0.008$ for both rewards). To conduct the mediation analyses, we used the CANlab Mediation toolbox available at https://github.com/canlab.

Of note, we performed a post-hoc power analysis to obtain an approximate estimate of the sample size that would be required to reach an 80% power to correctly detect each voxel tested as a significant brain mediator of the effect of bvFTD on discount rate or sensitivity to LL reward. For this purpose, we firstly estimated an approximate effect size for both path a and path b from the minimum absolute $Z$-values (using the following approximation: $r \approx \frac{Z}{\sqrt{(Z^2+df)}}$ with df = 36) found within significant clusters for path a and path b, respectively. For instance, for the discount rate for money, with Z-min = 0.04 within significant clusters for path a and Z-min = 3.67 within significant clusters for path b, we found that very small effect sizes were detected for path a ($r = 0.007$) while only large effect sizes were detected for path b ($r = 0.52$). We then used the reference table proposed by Fritz and MacKinnon[71] to estimate the required sample size for an 80% power in mediation analyses according to different parameters. We thus found that, in the case of small effect size for path a, large effect size for path b, and with the use of percentile bootstrapping, the required sample size was N = 398. This example serves to illustrate that our whole-brain mediation analyses (performed with N = 38 participants) are likely to be significantly underpowered, which diminishes the positive predictive value even of statistically significant findings[72].

## Reporting summary

Further information on research design is available in the Nature Portfolio Reporting Summary linked to this article.

## Data availability

Data are available on OSF. Structural MRI data and other clinical data are available from the corresponding author, upon reasonable request.

## Code availability

Scripts used for the ITC tasks and computation of ITC outcomes (discount rate and sensitivity to LL reward) are available on OSF.

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

## Acknowledgements

This study was funded by grant ANR-10-IAIHU-06 from the programme 'Investissements d'avenir', by grant FRM DEQ20150331725 from the foundation 'Fondation pour la recherche médicale', and by the ENEDIS company. This work was also funded by HP's Octapharma Chair in Decision Neuroscience and INSEAD's Research and Development Funds. We would like to thank patients, caregivers and organisers (in particular, Armelle Rametti-Lacroux) of the ECOCAPTURE consortium study and all the students who contributed to help data collection. We would also like to thank Raffaella Migliaccio for her advice on clinical details of the paper.

## Author contributions

Study conception and design: H.P., L.S., B.B., and R.L. Data acquisition: B.B. and A.D. Analysis and interpretation of data: V.G. and A.D. Study supervision: H.P., L.K., and V.G. Writing the first version of the manuscript: V.G. and H.P. Obtaining funding: H.P., R.L., and B.B. All authors critically revised the manuscript for its intellectual content and approved its final version.

## Competing interests

The authors declare no competing interests.
