## [Transparent Peer Review file · Communications Biology]

Mapping the brain atrophy mediating increased impatience for reward in frontotemporal dementia

Corresponding Author: Dr Valérie Godefroy

Version 0:

Reviewer comments:

Reviewer #1

(Remarks to the Author)

In this manuscript, Godefroy and colleagues investigate the neuroanatomical basis of altered delay discounting of monetary and food rewards (reward impatience) in patients with bvFTD. They observed higher delayed discounting (for both food and monetary rewards) in bvFTD patients compared to controls. Discounting rates correlated with symptoms related to disinhibition, changes in eating behavior, and executive apathy (measured via self-reports). Using whole-brain mediation analysis, they identified reduced grey matter in the medial pulvinar, parahippocampal cortex, and middle temporal lobe as brain mediator of altered delay discounting in bvFTD patients. The results of the study are interesting, especially due to the inclusion of primary (food) and secondary (money) rewards, the methods used are technically sound, and the manuscript is well-written. I have some methodological questions and comments that could improve the manuscript.

1.Regarding the computational modelling analyses, what is it meant with "inconsistent choices"? Were these choices made in a random fashion, or did they simply failed to fit a hyperbolic discounting model (and thus maybe a parabolic or linear model would better fit)? Was there a control condition (e.g. asking to identify which of the two choices would pay sooner than the other as in Chiong et al. 2015) to rule out that participants in the bvFTD sample were failing to understand the task structure or exhibiting a general comprehension or attentional deficit? Although I am aware that this would further reduce the statistical power due to the small sample, it would be still informative to conduct the analysis on the discount rate without the participants with inconsistent choices. Related to this, in the Discussion (p. 30 line 673-674) it is mentioned that "Of note, even without these approximations, delay discounting parameters were found higher in bvFTD patients than in controls.", but I could not find an analysis excluding the participants for whom the values were imputed in the manuscript or SM.

2.In the sensitivity analysis (Supp Fig. 1), did the outliers belong to the group of individuals for which values were imputed? And were they calculated on the whole sample or within each group? Also, the authors report that correlations between discount rate and sensitivity to LL reward were tested using Spearman correlations "after correcting for group effect." It would be helpful to clarify how this correction was performed. In general, it would be useful to better clarify in the manuscript how and for which analyses outliers were identified and excluded. It would be ideal to use the same approach for all analyses.

3.Correlations: it is confusing that some correlations are computed on the whole sample, and some additionally for each group separately. It is also unclear which correlation survived Bonferroni correction. The presence of significant group differences in both measures being correlated is expected to lead to a correlation. I recommend reporting only correlations conducted separately within each group and all corrected for multiple comparison (when examining the same hypothesis).

4.In the mediation analyses results, common regions among the 3 paths (a,b,ab) and the two types of rewards are identified and discussed. To robustly support claims of common mediation across paths and reward types, a conjunction analysis could be conducted.

Minor points:

- In the manuscript, the decision to impute the values is justified by the "strong linear relationship between %SS choice and estimated k. It would be useful to report the linear relationship coefficient (e.g. r value and/or a scatter plot).

- The statistics regarding the analyses shown in supp. Fig 2 is not reported neither in the text nor in the figure (expect for p

value). It would be useful to add it to the text.

•Please report also standard deviations and not only mean, median and range in text.

•Unless there are reasons not to, I think it would be useful to put the table describing sample characteristics in the manuscript, rather than in SM, please also add the age range.

Reviewer #2

(Remarks to the Author)

I have been interested to review this manuscript, which utilizes choice behavior in a structured experimental setting along with neuropsychological and caregiver report measures of impulsivity and structural MRI to investigate delay discounting for both food and monetary rewards in 22 people with bvFTD and 17 matched controls. The investigators report increased discounting for both types of rewards among participants with bvFTD (though see below regarding this interpretation). Behavior on this experimental task correlated with a formal neuropsychological measure of disinhibition and a caregiver report of eating behavior. Finally, the investigators report that the association between bvFTD diagnosis and delay discounting (again, see below) is mediated across both tasks by gray matter atrophy in the left medial pulvinar thalamic nucleus, the left parahippocampal cortex, and the right middle temporal cortex.

This study has several strengths, but there are also several concerns that diminish my enthusiasm for the manuscript in its present form.

Major concerns:

1 - A key innovation claimed by the authors is the use of a mediation analysis rather than more conventional correlational analyses. However, the investigators do not address some of the classic statistical challenges involved in tests of mediation, and the lack of documentation makes it difficult to assess the positive predictive value of their findings. In general, the use of mediation can exacerbate what is widely recognized as a problem of low power and therefore low reliability in small-n neuroimaging studies. A key reference is Fritz & MacKinnon, *Psychol Sci* 2007, who show that the required sample size of a mediation analysis depends greatly on the expected size of the mediation effect. The most pessimistic assumptions (small α , small β , $\tau = 0$) yield a required sample size of 20,886; whereas even the most optimistic assumptions (large α , large β , $\tau = 0.59$) yield a required sample size of 36, only 2 fewer than the sample in this paper. Unless the authors have reason to expect the only most optimistic assumptions for all three parameters to hold, this study is likely to be significantly underpowered, which markedly diminishes the positive predictive value even of a statistically significant finding. (Button et al, *Nat Rev Neurosci* 2013) Ideally, a formal a priori power analysis would have been conducted and preregistered at the outset of any mediation analysis given the extreme sensitivity of sample size calculations in mediation analysis to initial assumptions, but even a post hoc power analysis would assist the reader in assessing the credibility of the assumptions underlying the project.

2 - Because behavior on the experimental tasks correlates with a formal neuropsychological measure of disinhibition and a caregiver report measure of impulsive eating, the investigators propose that their findings may yield transdiagnostic neural markers of impatience and more accurate phenotyping of impulsivity symptoms. These claims are disconnected from key work on transdiagnostic associations and on impulsivity. Related to the transdiagnostic claim, there is no basis offered here for claiming that results from one patient group can generalize to other clinical groups that manifest impatience, and the reliance on anatomic correlations from within a single group yields the well-documented co-atrophy problem (Sollberger et al, *Neuropsychologia* 2009). Meanwhile, the claims in the paper regarding impulsivity seem to treat impulsivity as a unidimensional construct, whereas extensive work shows impulsivity to encompass several distinct facets or dimensions. See, for example, the UPPS (Urgency, Premeditation, Perseverance, Sensation seeking) model.

3 - I have serious concerns about the use of the hyperbolic discounting model given acknowledged problems of model fit. The hyperbolic discounting model characterizes intertemporal choice as falling along a single dimension characterized by the discount rate k , and has been used to differentiate temporal attitudes in healthy populations. However, to apply this model to people with bvFTD assumes that differences in behavior between healthy and disease populations must fall along the same dimension as variation in behavior among the healthy population; i.e., that pathological changes in bvFTD affect the discount parameter k rather than the shape of the discounting function itself. This assumption is in fact violated; 5 bvFTD participants in the money paradigm and 6 in the food paradigm gave responses that could not be fit to the hyperbolic model, whereas none of the healthy participants had such patterns. Despite the failure of the model to fit the data, the investigators attempted to still apply the hyperbolic model and discount rate to characterize choices in bvFTD by linearly interpolating k values based on the SS%. Considering the 7 bvFTD "mono-choice" patterns in the money paradigm and 5 in the food paradigm, this means that a majority of the data from the bvFTD group in both tasks could not be expressed in terms of a computable k value and was instead used to impute a k value. Since the majority of the data from the bvFTD group cannot be characterized using the hyperbolic discounting model, this raises some questions about whether altered intertemporal choice in this disease should be analyzed using a delay discounting framework. The authors write that Supplementary Figure 2 shows that bvFTD participants showed lower sensitivity to LL rewards than controls; however, examining Supplementary Figure 2 the median beta value for LL reward magnitude does not differ from zero. This suggests that bvFTD participants are not applying a "discount" to later rewards, but are instead ignoring them. I would argue that given clear violations of the assumptions of the discounting framework, a less assumption-laden framework for characterizing altered intertemporal choice in bvFTD should be considered. (This also exacerbates concerns about whether findings in bvFTD

generalize transdiagnostically to other conditions in which delay discounting is magnified.)

Minor concerns:

1 - The paper title, "Topographical distribution of structural impairments mediating increased impatience for reward," is unnecessarily opaque and confusing. The term "topography" in neuroscience and neuroanatomy typically refers to maps such as V1 or the sensory homunculus in which there is an ordered spatial relationship between brain structures underlying a representation and spatial or other features among what is represented. I don't see evidence for such a relationship here. Also, I do not understand the referent of the term, "structural impairment."

2 - Given the known association of medial pulvinar atrophy with the C9orf72 mutation, do the authors have genotyping data for the participants in this study? Or if not, can the bvFTD cohort be characterized in terms of the proportion of familial vs. sporadic cases?

3 - The investigators note that previous studies of similar size have not demonstrated anatomical associations with greater impatience in bvFTD. This state of affairs would establish a prior that there is no such anatomical association or that any such association is small. (See major concern 1 above.) How do the authors explain the discordance between their positive finding and these previously reported negative findings?

Version 1:

Reviewer comments:

Reviewer #1

(Remarks to the Author)

I thank the authors for thoroughly addressing my comments and revising their manuscript. I am satisfied with the changes made and have no further comments.

Point-by-point responses to reviewer comments

Reviewer 1

In this manuscript, Godefroy and colleagues investigate the neuroanatomical basis of altered delay discounting of monetary and food rewards (reward impatience) in patients with bvFTD. They observed higher delayed discounting (for both food and monetary rewards) in bvFTD patients compared to controls. Discounting rates correlated with symptoms related to disinhibition, changes in eating behavior, and executive apathy (measured via self-reports). Using whole-brain mediation analysis, they identified reduced grey matter in the medial pulvinar, parahippocampal cortex, and middle temporal lobe as brain mediator of altered delay discounting in bvFTD patients. The results of the study are interesting, especially due to the inclusion of primary (food) and secondary (money) rewards, the methods used are technically sound, and the manuscript is well-written. I have some methodological questions and comments that could improve the manuscript.

Response: We thank reviewer 1 for underlining the strengths of our study. Reviewer 1's methodological comments have been addressed by: 1. adding clarifications regarding the analyses that had already been performed (e.g., clarifications about inconsistent choices, removed outliers); 2. adding new analyses using the intertemporal choice outcome of sensitivity to larger later reward amount, which is not based on a model of hyperbolic discounting (and thus did not require any approximation like the discount rate); 3. replacing correlation tests previously performed across the whole sample (bvFTD patients and controls) by correlation tests within the patient sample to avoid that group effect drives the correlation; 4. adding conjunction analyses for the whole-brain mediations to identify common clusters across paths a, b and ab.

1. Regarding the computational modelling analyses, what is it meant with "inconsistent choices"? Were these choices made in a random fashion, or did they simply failed to fit a hyperbolic discounting model (and thus maybe a parabolic or linear model would better fit)? Was there a control condition (e.g. asking to identify which of the two choices would pay sooner than the other as in Chiong et al. 2015) to rule out that participants in the bvFTD sample were failing to understand the task structure or exhibiting a general comprehension or attentional deficit? Although I am aware that this would further reduce the statistical power due to the small sample, it would be still informative to conduct the analysis on the discount rate without the participants with inconsistent choices. Related to this, in the Discussion (p. 30 line 673-674) it is mentioned that "Of note, even without these approximations, delay discounting parameters were found higher in bvFTD patients than in controls.", but I could not find an analysis excluding the participants for whom the values were imputed in the manuscript or SM.

Response: To address this comment, we have provided further methodological details regarding the intertemporal choice tasks and computation of discount rates, in particular the definition of inconsistent patterns of choice. We have also added to the Supplementary Material the results of comparisons between groups for the discount rates after removing the cases of inconsistent choices that had been approximated. Moreover, we have added new results using a second outcome of intertemporal choice tasks: the sensitivity to larger later reward amounts (i.e., the extent to which the larger later reward amount impacted the participant's choice between the SS and LL options). This second outcome did not require any approximation and allowed us to further support the results obtained with the discount rate outcome.

Inconsistent patterns of choice are now clearly defined in the manuscript (on lines 337-338): “inconsistent patterns of answers (i.e., with the probability of choosing the LL option decreasing with larger amounts of LL rewards)” [...] “For these participants showing patterns of unique answers or very incoherent answers, the discounting model did not fit and we could not use the same method of estimation.” Indeed, these participants did not apply any discount to the LL reward option, which is why we could not use any discounting model and we had to estimate their discount rate from the percentage of choice of SS option (%SS values).

Regarding the control condition for the intertemporal choice tasks, we have added on lines 218-220 that: “There was a test trial at the beginning of each task to verify participant’s correct understanding and an experimenter was present to address potential questions throughout the task (especially for bvFTD patients).”

In the revised version of the Supplementary files, we have added the “Effect of bvFTD on discount rate after removing incoherent participants.” in Supplementary Figure 1 to show that “Results remain globally similar after removing incoherent participants, i.e., discount rates are higher in bvFTD patients compared to controls (with a close-to-significant effect for Money and a significant effect for Food).”

Finally, instead of performing all the analyses on the discount rate without the participants with inconsistent choices (and thus losing statistical power), we have added to the manuscript new analyses using the parameter of sensitivity to LL reward showing that: 1. it is lower in bvFTD patients than in controls (see Figure 2); 2. it correlates significantly with lower executive functions and close-to-significantly with inhibition deficits among patients (see Figure 3); 3. its decrease in bvFTD is mediated by lower grey matter density in medial pulvinar and parahippocampal cortex, like the discount rate (see Figures 4, 5 and Supplementary Figure 3). Thus, we provided a second marker of impatience for reward in bvFTD which did not require any approximation and confirmed the robustness of results. This second marker indeed supported the findings obtained with the discount rate regarding the neuroanatomical modifications underlying increased impatience for reward in bvFTD.

2. In the sensitivity analysis (Supp Fig. 1), did the outliers belong to the group of individuals for which values were imputed? And were they calculated on the whole sample or within each group? Also, the authors report that correlations between discount rate and sensitivity to LL reward were tested using Spearman correlations “after correcting for group effect.” It would be helpful to clarify how this correction was performed. In general, it would be useful to better clarify in the manuscript how and for which analyses outliers were identified and excluded. It would be ideal to use the same approach for all analyses.

Response: Thank you for these suggestions. Firstly, to simplify the question of outliers, in the revised version of the manuscript, we have systematically removed only the extreme outliers calculated on the whole sample for the intertemporal choice task outcomes (discount rate and sensitivity to LL reward amount). Secondly, we have clarified the method used to correct for group effect when testing the relationship between the discount rate and sensitivity to LL reward.

In the revised manuscript, we have provided the following details (see lines 355-360): “To avoid biasing correlation and mediation analyses, extreme outliers (i.e., $< Q1 - 3*(Q3-Q1)$ or $> Q3 + 3*(Q3-Q1)$) identified within the whole sample were systematically removed for both the computed discount rates and sensitivities to larger later rewards. Thus, two participants (in control group) were removed for the discount rate for food and three other participants (two in control group and one in patient group) were removed for the sensitivity to LL reward for food.” Of note, removed outliers did not belong to the group of individuals for whom values of discount rates were imputed.

Moreover, for both the money and food paradigms, to check the validity of our imputation method for the computation of discount rates, we verified that the participants' discount rates were related to how sensitive they were to LL reward. We have now clarified this in the Method section (see lines 362-370): "Using nonparametric Spearman rank correlations, we tested the links between the computed discount rates and sensitivity to LL reward after correction for group effect to verify that higher discount rates were associated with lower sensitivity to LL reward. More precisely, we tested the correlations between the residuals of the linear regression predicting discount rates from group (bvFTD vs. control) on the one hand and the residuals of the linear regression predicting sensitivity to LL rewards from group (bvFTD vs. control) on the other hand. These residuals account for the part of the variability in the scores of discount rate and sensitivity to LL reward that is not due to group effect."

3. Correlations: it is confusing that some correlations are computed on the whole sample, and some additionally for each group separately. It is also unclear which correlation survived Bonferroni correction. The presence of significant group differences in both measures being correlated is expected to lead to a correlation. I recommend reporting only correlations conducted separately within each group and all corrected for multiple comparison (when examining the same hypothesis).

Response: We agree with these remarks regarding correlation tests. In the revised manuscript, we have expanded the correlations tested between potential markers of impatience for reward and bvFTD symptoms (all the symptoms potentially involving the urgency facet of impulsivity). Moreover, we have only reported correlations tested within the patient group (as symptom measures are mostly relevant in this group and less among controls) to avoid that group differences drive the observed correlations. We have also added Bonferroni corrections for multiple correlation tests for each of the two complementary outcomes of the intertemporal choice task (discount rate and sensitivity to LL reward).

Our hypotheses on correlations are detailed in the introduction (see lines 133-141): "we assumed that, as markers of impatience for reward in bvFTD, delay discounting outcomes should correlate with symptoms of bvFTD that are related to impulsivity (especially its urgency component): (1) disinhibition (or deficit of inhibition), which can be considered as a preference for the most immediate and prepotent answers across various contexts (e.g., preference for immediate reactions of hostility and aggressiveness when confronted to frustration or irritation), (2) deficits of executive functions, which include lack of inhibitory and attentional control and (3) eating behaviour changes such as binge eating and preference for sweet foods, which may also correspond to the expression of a preference for immediate rewards."

Details regarding correlation analyses are in the Method section (see lines 381-390): "Our hypothesis was that symptoms of inhibition deficits, lower executive functions and eating behaviour changes were related to higher discount rate and lower sensitivity to LL reward in bvFTD patients. We used non-parametric Spearman rank correlations to test the links between the participants' discount rates ($\log(k)$) for money and for food rewards and: (1) the Hayling error score (measuring inhibition deficit), (2) the FAB (measuring executive functions) and (3) the EBI total score (measuring eating behaviour changes). We also tested the links with the executive apathy subtype (not with the initiation and emotional apathy subtypes) since executive apathy is conceptually related to executive dysfunctions. We tested all these associations within the group of bvFTD patients. The exact same correlation tests were performed with the sensitivity to LL reward for money and food rewards."

In the result section (see lines 488-492), we have also added that: "For each of the two ITC task outcomes, we tested correlations with four symptoms suggestive of impulsivity among bvFTD patients: inhibition deficit, lower executive functions, eating behaviour changes (such as binge eating), and executive apathy (or apathy due to executive dysfunction), for both money

and food rewards (leading to a total of 8 tested correlations per ITC outcome). We applied a Bonferroni correction to control for multiple testing of correlations for each ITC outcome.”

Two results remained significant after Bonferroni correction (see lines 516-520): “Among patients, higher discount rates are closely related to inhibition deficits while lower sensitivities to LL reward are strongly associated with impaired executive functions.” Thus, “results show that the discount rate and sensitivity to LL reward for monetary rewards can be considered as markers of individual differences of impatience, related to impulsivity symptoms, among bvFTD patients.”

4. In the mediation analyses results, common regions among the 3 paths (a,b,ab) and the two types of rewards are identified and discussed. To robustly support claims of common mediation across paths and reward types, a conjunction analysis could be conducted.

Response: We are grateful for this suggestion which has allowed us to strengthen the robustness of our conclusions. In the revised version, we have added conjunction analyses to identify the common regions between the three paths (a, b and ab) for the two whole-brain mediation analyses conducted with the two confirmed markers of impatience for reward among bvFTD patients. Moreover, we computed the intersection of the resulting maps to identify the common mediators of increased discount rate and decreased sensitivity to LL reward for money rewards in bvFTD.

Results of these conjunction analyses are shown in the revised Supplementary files in Supplementary Figure 3 and they are mentioned in the revised manuscript as follows (see lines 567-580): “The intersection of voxels with significant paths a, b, and ab was then interpreted as a set of mediating brain regions (see results of the conjunction analyses showing intersections between paths for discount rate in Supplementary Figure 3.A and for sensitivity to LL reward in Supplementary Figure 3.B). Among these mediating regions, some were detected in similar locations for both the discount rate and the sensitivity to LL reward with money: mostly in the left medial pulvinar thalamic nucleus and to a lesser extent, in the left parahippocampal cortex (see results of the conjunction analysis showing the intersection between mediating brain regions for discount rate and mediating brain regions for sensitivity to LL reward in Supplementary Figure 3.C). In these regions, bvFTD patients presented significant grey matter atrophy (path a), lower GMD was associated with higher discount rate and lower sensitivity to LL reward (path b), and the loss of GMD due to bvFTD contributed to increase the discount rate and decrease the sensitivity to LL reward in patients (path ab).”

Minor points:

- In the manuscript, the decision to impute the values is justified by the “strong linear relationship between %SS choice and estimated k. It would be useful to report the linear relationship coefficient (e.g. r value and/or a scatter plot).

Response: To address this comment, we have reported the correlation coefficients which indicate strong linear relationships (according to the guide by Evans (1996) on correlation size) between %SS choice and estimated k’s. See lines 341-343: “We took advantage of the strong linear relationship between %SS choice and estimated k’s (for monetary rewards, $R = 0.64$; $p = 0.001$ and for food rewards, $R = 0.69$; $p < 0.001$)”.

- The statistics regarding the analyses shown in supp. Fig 2 is not reported neither in the text nor in the figure (except for p value). It would be useful to add it to the text.

Response: In the revised version, these results are now displayed in the main manuscript (see lines 456-459) with all the statistics: “bvFTD patients showed lower sensitivity to LL rewards than controls for both money ($W=81.5$, $p=0.005$, effect size=0.46) (see Fig. 2.C) and food rewards ($W=61$, $p=0.003$, effect size=0.51) (see Fig. 2.D).”

- Please report also standard deviations and not only mean, median and range in text.

Response: In the result section in the paragraph from line 433 to 445, we have reported all the standard deviations in addition to other statistics for both the discount rate and the sensitivity to LL reward, for money and food rewards.

- Unless there are reasons not to, I think it would be useful to put the table describing sample characteristics in the manuscript, rather than in SM, please also add the age range.

Response: In the revised version, the table describing the sample characteristics is in the main manuscript as “Table 1. Demographic and main clinical measures of bvFTD patients and controls.” (see lines 195-206) and we have added the age for both groups.

Reviewer 2

I have been interested to review this manuscript, which utilizes choice behavior in a structured experimental setting along with neuropsychological and caregiver report measures of impulsivity and structural MRI to investigate delay discounting for both food and monetary rewards in 22 people with bvFTD and 17 matched controls. The investigators report increased discounting for both types of rewards among participants with bvFTD (though see below regarding this interpretation). Behavior on this experimental task correlated with a formal neuropsychological measure of disinhibition and a caregiver report of eating behavior. Finally, the investigators report that the association between bvFTD diagnosis and delay discounting (again, see below) is mediated across both tasks by gray matter atrophy in the left medial pulvinar thalamic nucleus, the left parahippocampal cortex, and the right middle temporal cortex.

This study has several strengths, but there are also several concerns that diminish my enthusiasm for the manuscript in its present form.

Response: We thank reviewer 2 for providing detailed and insightful feedbacks. Reviewer 2's concerns have been addressed mainly by: 1. adding a post-hoc power analysis and discussing related limitations; 2. toning down the transdiagnostic claim; 3. performing all the analyses with the sensitivity to larger later reward amount, a complementary outcome from the intertemporal choice tasks that is less assumption-laden than the discounting rate.

Major concerns:

1 - A key innovation claimed by the authors is the use of a mediation analysis rather than more conventional correlational analyses. However, the investigators do not address some of the classic statistical challenges involved in tests of mediation, and the lack of documentation makes it difficult to assess the positive predictive value of their findings. In general, the use of mediation can exacerbate what is widely recognized as a problem of low power and therefore low reliability in small-n neuroimaging studies. A key reference is Fritz & MacKinnon, *Psychol Sci* 2007, who show that the required sample size of a mediation analysis depends greatly on the expected size of the mediation effect. The most pessimistic assumptions (small α , small β , $r = 0$) yield a required sample size of 20,886; whereas even the most optimistic assumptions (large α , large β , $r = 0.59$) yield a required sample size of 36, only 2 fewer than the sample in this paper. Unless the authors have reason to expect the only most optimistic assumptions for all three parameters to hold, this study is likely to be significantly underpowered, which markedly diminishes the positive predictive value even of a statistically significant finding. (Button et al, *Nat Rev Neurosci* 2013) Ideally, a formal a priori power analysis would have been conducted and preregistered at the outset of any mediation analysis given the extreme sensitivity of sample size calculations in mediation analysis to initial assumptions, but even a post hoc power analysis would assist the reader in assessing the credibility of the assumptions underlying the project.

Response: We thank Reviewer #2 for this insightful comment and suggestion of post-hoc power analysis. We have added to the revised manuscript a post-hoc power analysis: we used the observed minimum effect sizes detected as significant for path a and path b to determine the sample size that would have been required to obtain an 80% power for our whole-brain mediation analyses. The required sample size was $N \approx 398$, which is very hard to reach especially for a population of patients with a rather rare disease. We have thus stressed again the issue of lack of statistical power in the discussion of limits.

In the method section (in subsection 2.5.4 describing whole-brain mediations), we have added the following paragraph (see lines 412-427): "Of note, we performed a post-hoc power analysis to obtain an approximate estimate of the sample size that would be required to reach an 80% power to correctly detect each voxel tested as a significant brain mediator of the effect of bvFTD on discount rate or sensitivity to LL reward. For this purpose, we firstly estimated an approximate effect size for both path a and path b from the minimum absolute Z-values (using

the following approximation: $r \approx Z / (\sqrt{Z^2 + df})$ with $df=36$) found within significant clusters for path a and path b respectively. For instance, for the discount rate for money, with $Z_{\min}=0.04$ within significant clusters for path a and $Z_{\min}=3.67$ within significant clusters for path b, we found that very small effect sizes were detected for path a ($r \approx 0.007$) while only large effect sizes were detected for path b ($r \approx 0.52$). We then used the reference table proposed by Fritz and MacKinnon⁴⁶ to estimate the required sample size for an 80% power in mediation analyses according to different parameters. We thus found that, in the case of small effect size for path a, large effect size for path b, and with the use of percentile bootstrapping, the required sample size was $N \approx 398$. This example serves to illustrate that our whole-brain mediation analyses (performed with $N=38$ participants) are likely to be significantly underpowered, which diminishes the positive predictive value even of statistically significant findings⁴⁷.”

We have also added the following remarks to the discussion (see lines 724-732): “The small sample size leads to a substantial lack of statistical power for the mediation analyses, which undermines the reliability of these results. Our sample size of bvFTD patients is however very close to the upper limit of the range of sample sizes used in previous studies of delay discounting in bvFTD (i.e., between $N=14$ and $N=28$). Moreover, we could still identify significant clusters with an interesting convergence of results between two different frameworks used to characterize altered intertemporal preferences in bvFTD: the assumption-laden outcome of discount rate and the less assumption-laden outcome (not driven by any hypothesis of hyperbolic discounting) of sensitivity to larger later reward.”

2 - Because behavior on the experimental tasks correlates with a formal neuropsychological measure of disinhibition and a caregiver report measure of impulsive eating, the investigators propose that their findings may yield transdiagnostic neural markers of impatience and more accurate phenotyping of impulsivity symptoms. These claims are disconnected from key work on transdiagnostic associations and on impulsivity. Related to the transdiagnostic claim, there is no basis offered here for claiming that results from one patient group can generalize to other clinical groups that manifest impatience, and the reliance on anatomic correlations from within a single group yields the well-documented co-atrophy problem (Sollberger et al, *Neuropsychologia* 2009). Meanwhile, the claims in the paper regarding impulsivity seem to treat impulsivity as a unidimensional construct, whereas extensive work shows impulsivity to encompass several distinct facets or dimensions. See, for example, the UPPS (Urgency, Premeditation, Perseverance, Sensation seeking) model.

Response: To address this comment, we have toned down the transdiagnostic claim in the discussion and simply suggested that lesions in the regions identified as common mediators of the two markers of higher impatience for reward in bvFTD (that is increased discount rates and decreased sensitivity to LL reward for money) might constitute targets of interest for the early detection and treatment of impulsivity symptoms involving urgency. Moreover, we have specified in the introduction that we expected a link between potential markers of impatience for reward (from the ITC tasks) and bvFTD symptoms involving the Urgency facet of impulsivity (according to the UPPS model).

In the discussion, we have been very clear about the fact that our findings concern bvFTD patients and not all patients showing impulsivity and impatience for reward (e.g., “In conclusion, this study investigated the relationships between markers of impatience for reward, symptoms and brain structure in the neuropathological condition of bvFTD. Key findings suggest that in bvFTD: [...]” on lines 754-756).

In the discussion of limits (see lines 733-741), we have stressed that: “clusters evidenced as mediators (in particular, the medial pulvinar and parahippocampal cortex, according to the results of the conjunction analysis between paths a, b and ab) were also regions for which the effect was independent of bvFTD group effect on brain atrophy. This is supposed to prevent the well-documented co-atrophy issue⁷¹ and suggests a possible transdiagnostic value of our results.”

Thus, we have only added the following future perspectives to consider in the conclusion (see lines 765-776): “In terms of translational impacts, our results suggest that the delay discounting task provides markers of inhibition deficits and executive disorders and could thus potentially contribute to a better phenotyping of conditions associated with a marked impatience for reward. They also suggest that the anatomy of specific brain regions such as the medial pulvinar and parahippocampal cortex may constitute targets of interest for the early detection and treatment of impulsivity symptoms involving urgency.” However, we have added that: “one of the main limitations of our results is that they only concern bvFTD condition and need to be further validated across a broader panel of neurodegenerative and psychiatric conditions.”

Finally, in the introduction (see lines 100-110), we have specified that: “The main clinical symptoms observed in bvFTD patients are disinhibition, apathy, loss of empathy, perseverative, stereotyped or compulsive behaviour, eating behaviour changes and executive deficits⁸. [...] Higher impatience for reward might be a core common factor linked to several of these symptoms⁶. Therefore, identifying the neuroanatomical correlates of this impatience in bvFTD may have translational implications. It could in particular inform the treatment of potentially related symptoms that involve the urgency facet of impulsivity, defined as the tendency to act rashly especially in an emotional context according to the UPPS (Urgency, Premeditation, Perseverance, Sensation seeking) model of impulsivity²².”

3 - I have serious concerns about the use of the hyperbolic discounting model given acknowledged problems of model fit. The hyperbolic discounting model characterizes intertemporal choice as falling along a single dimension characterized by the discount rate k , and has been used to differentiate temporal attitudes in healthy populations. However, to apply this model to people with bvFTD assumes that differences in behavior between healthy and disease populations must fall along the same dimension as variation in behavior among the healthy population; i.e., that pathological changes in bvFTD affect the discount parameter k rather than the shape of the discounting function itself. This assumption is in fact violated; 5 bvFTD participants in the money paradigm and 6 in the food paradigm gave responses that could not be fit to the hyperbolic model, whereas none of the healthy participants had such patterns. Despite the failure of the model to fit the data, the investigators attempted to still apply the hyperbolic model and discount rate to characterize choices in bvFTD by linearly interpolating k values based on the SS%. Considering the 7 bvFTD "mono-choice" patterns in the money paradigm and 5 in the food paradigm, this means that a majority of the data from the bvFTD group in both tasks could not be expressed in terms of a computable k value and was instead used to impute a k value. Since the majority of the data from the bvFTD group cannot be characterized using the hyperbolic discounting model, this raises some questions about whether altered intertemporal choice in this disease should be analyzed using a delay discounting framework. The authors write that Supplementary Figure 2 shows that bvFTD participants showed lower sensitivity to LL rewards than controls; however, examining Supplementary Figure 2 the median beta value for LL reward magnitude does not differ from zero. This suggests that bvFTD participants are not applying a *discount* to later rewards, but are instead ignoring them. I would argue that given clear violations of the assumptions of the discounting framework, a less assumption-laden framework for characterizing altered intertemporal choice in bvFTD should be considered. (This also exacerbates concerns about whether findings in bvFTD generalize transdiagnostically to other conditions in which delay discounting is magnified.)

Response: We thank Reviewer #2 for underlining these issues regarding the estimation of discount rates in bvFTD to quantify their level of impatience for reward (or tendency to prefer smaller sooner to larger later rewards). As suggested by Reviewer #2, we have considered a less assumption-laden framework for characterizing altered intertemporal choice in bvFTD and in the revised manuscript, we have performed all the analyses (including correlations with symptoms and whole-brain mediation) with a complementary outcome of the intertemporal choice task: the sensitivity to LL reward amount (or the impact of LL reward amount on the choice between SS and LL options). This outcome can be computed independently of the delay

discounting framework as it is not based on the assumption that reward subjective value decreases with time. Estimating this outcome thus did not require any approximation. Like the discount rate, this outcome is supposed to be related to impatience for reward: higher impatience implies a tendency to more systematically prefer the SS option regardless of LL reward amount and therefore a lower sensitivity to LL reward amount.

At the end of the introduction of the revised manuscript (see lines 154-161), we explain that: “we estimated two potential markers of impatience for reward: the discount rate k and the sensitivity to larger later reward amount (i.e., the extent to which the larger later reward amount impacts the participant’s choice, which is assumed to be lower with higher impatience for reward). We compared these delay discounting outcomes between patients and controls, and tested their links with bvFTD symptoms. Finally, using the statistical framework of whole-brain mediation analysis^{32,33} applied to structural MRI, we aimed at identifying the brain areas in which structural differences between the patients and healthy controls explained group differences in markers of impatience for reward.” Accordingly, we now refer to these two outcomes of intertemporal choice tasks (instead of mentioning exclusively the discount rate), as measuring “intertemporal preferences” or “preference for short-term rewards” or “impatience for reward” throughout the manuscript.

In the method section (see lines 348-354), we have detailed how this second outcome was computed: “For each participant and each reward type, we also computed the individual sensitivity to larger later (LL) reward as an outcome that does not rely on an *a priori* assumption of hyperbolic discounting, to complement the discount rates. For this purpose, we fitted a logistic regression model in each participant predicting the trial-to-trial probability of choosing the LL option from the LL reward amount and LL delay value. We used the regression coefficient of LL reward amount as an estimate of the individual sensitivity to LL reward (higher sensitivity to reward corresponding to higher values).”

Using sensitivity to LL reward amount allowed us to confirm that, like the discount rate, this outcome with money rewards could be used as a marker of impatience for reward among bvFTD patients (“Together, results show that the discount rate and sensitivity to LL reward for monetary rewards can be considered as markers of individual differences of impatience, related to impulsivity symptoms, among bvFTD patients. Among patients, higher discount rates are closely related to inhibition deficits while lower sensitivities to LL reward are strongly associated with impaired executive functions.” on lines 517-521). Moreover, it allowed us to reinforce the robustness of our neuroanatomical results: indeed, overlapping clusters in the medial pulvinar and parahippocampal cortex have been found as mediators of both the increase in discount rate and the decrease in sensitivity to LL reward in bvFTD. This is detailed in the result section (see lines 570-578, Figures 4-5 and Supplementary Figure 3): “Among these mediating regions, some were detected in similar locations for both the discount rate and the sensitivity to LL reward with money: mostly in the left medial pulvinar thalamic nucleus and to a lesser extent, in the left parahippocampal cortex (see results of the conjunction analysis showing the intersection between mediating brain regions for discount rate and mediating brain regions for sensitivity to LL reward in Supplementary Figure 3.C). In these regions, bvFTD patients presented significant grey matter atrophy (path a), lower GMD was associated with higher discount rate and lower sensitivity to LL reward (path b), and the loss of GMD due to bvFTD contributed to increase the discount rate and decrease the sensitivity to LL reward in patients (path ab).”

Despite the limitations underlined by Reviewer #2 and mentioned in the discussion of our manuscript (see lines 743-753), we kept the analyses involving the discount rate as a potential marker of impatience for reward. At least two results support the validity of this outcome, in addition to the sensitivity to LL reward: 1/ the significant correlation between discount rates and sensitivity to LL reward independently of group effect; 2/ the significant correlation between the discount rate for money and inhibition deficits among bvFTD patients, which confirms the link between this ITC outcome and the urgency facet of impulsivity (or the tendency to act in a rash, without properly considering consequences). Moreover, the fact that bvFTD patients tend to

ignore the LL option is not contradictory with the discounting framework. Indeed, this can be due to a potential underlying mechanism of delay discounting (proposed by Frost and McNaughton – see lines 645-648), which is the loss of salience of reward with the perceived distance to reach it (due to delay). If this loss of salience is extreme, this can lead to a tendency to ignore the LL option (lines 648-651: “Results indeed suggest that bvFTD patients act as if they tended to ignore the less salient later options and were biased in favour of the most salient smaller but sooner options, thus confirming previous observations in bvFTD”). Of note, the median value of sensitivity to LL reward is close to 0 because some patients display negative values of sensitivity to LL reward (corresponding to incoherent patterns of choice that increase the probability of choosing the LL option with lower LL reward amounts) but most patients show small but positive values, as expected according to the discounting framework.

Minor concerns:

1 - The paper title, "Topographical distribution of structural impairments mediating increased impatience for reward," is unnecessarily opaque and confusing. The term "topography" in neuroscience and neuroanatomy typically refers to maps such as V1 or the sensory homunculus in which there is an ordered spatial relationship between brain structures underlying a representation and spatial or other features among what is represented. I don't see evidence for such a relationship here. Also, I do not understand the referent of the term, "structural impairment."

Response: We have revised the paper title to make it less confusing. The new title is: "Mapping the brain atrophy mediating increased impatience for reward in frontotemporal dementia".

2 - Given the known association of medial pulvinar atrophy with the C9orf72 mutation, do the authors have genotyping data for the participants in this study? Or if not, can the bvFTD cohort be characterized in terms of the proportion of familial vs. sporadic cases?

Response: Thank you for this suggestion. We have added this precision to the paragraph describing bvFTD sample in the methods (see lines 192-193): "Among the 22 bvFTD patients included in the analyses, five patients (22.7% of the patient sample) had a pathological expansion in the C9 open reading frame 72 gene."

3 - The investigators note that previous studies of similar size have not demonstrated anatomical associations with greater impatience in bvFTD. This state of affairs would establish a prior that there is no such anatomical association or that any such association is small. (See major concern 1 above.) How do the authors explain the discordance between their positive finding and these previously reported negative findings?

Response: To address this comment, we have further explained in the introduction why the two previous studies were not adapted to demonstrate anatomical associations with greater impatience in bvFTD. See lines 118-121: "these studies were not designed for the specific purpose of studying delay discounting in bvFTD. One of these studies failed to find significant correlations between brain atrophy and discounting rate probably because they did not manage to evidence between-group differences at the behavioral level¹⁵." And on lines 125-130 regarding the second study: "this study pooled AD patients, bvFTD patients and controls for the study of the neuroanatomical correlates of discount rates, which was not adapted to establish a pattern of brain structural changes leading to increased discount rates specifically in bvFTD. Besides, their main finding that increased discount rates were associated with greater bilateral amygdala atrophy was driven by AD patients and not by bvFTD patients."